# Earth Stewardship Science—Transdisciplinary Contributions to Quantifying Natural and Cultural Heritage of Southernmost Africa

**Bastien Linol \*,†, Warren Miller †, Cheryl Rensburg †, Renee Schoeman †, Lucian Bezuidenhout †, Fabien Genin †, Barry Morkel †, Nyaradzo Dhliwayo †, Keegan Jeppesen †, Sinazo Dlakavu †, Okuhle Poto †, Gaathier Mahed †, Natasha Gariremo †, James Berkland †, Debbie Claassen †, Tebogo Sebake †, Thulisile Kunjwa †, Gregorio Guzzo †, Maria Bobbio †, Romain Pellen †, Lizalise Mngcele †, Manyano Makuzeni †, Katherine Winkler †, Thandeka Tembe †, Sameera Musa †, Khaya Valashiya †, Vhuhwavhohau Nengovhela †, Verouschka Sonn †, Divan Stroebel †, Nokuthua Kom †, Philasande Mambalu †, Taufeeq Dhansay †, Thomas Muedi † and Thakane Ntholi †**

AEON—Africa Earth Observatory Network, Nelson Mandela University, Summerstrand, Port Elizabeth 6031, South Africa; s206027541@mandela.ac.za (W.M.); cheryl.rensburg@mandela.ac.za (C.R.); s214234177@mandela.ac.za (R.S.); s207074356@mandela.ac.za (L.B.); s219869235@mandela.ac.za (F.G.); Barry.Morkel@mandela.ac.za (B.M.); Nyaradzo.Dhliwayo@mandela.ac.za (N.D.); s210042265@mandela.ac.za (K.J.); s214391396@mandela.ac.za (S.D.); s217699626@mandela.ac.za (O.P.); Gaathier.Mahed@mandela.ac.za (G.M.); s215319621@mandela.ac.za (N.G.); s213244233@mandela.ac.za (J.B.); dkilian@geoscience.org.za (D.C.); s213296748@mandela.ac.za (T.S.); s215122739@mandela.ac.za (T.K.); s220019835@mandela.ac.za (G.G.); s220019819@mandela.ac.za (M.B.); s221605673@mandela.ac.za (R.P.); s210050128@mandela.ac.za (L.M.); s208001229@mandela.ac.za (M.M.); s220042209@mandela.ac.za (K.W.); Thandeka.Tembe@mandela.ac.za (T.T.); s215027132@mandela.ac.za (S.M.); s214221008@mandela.ac.za (K.V.); s207043616@live.nmmu.ac.za (V.N.); s208091252@mandela.ac.za (V.S.); Divan.Stroebel@mandela.ac.za (D.S.); 200908211@ufh.ac.za (N.K.); s217074030@mandela.ac.za (P.M.); s211276863@mandela.ac.za (T.D.); s211266973@mandela.ac.za (T.M.); s212469843@mandela.ac.za (T.N.)

**\*** Correspondence: bastien.linol@mandela.ac.za

**†** All authors belong to AEON's Earth Stewardship Research Group.

**Abstract:** Evaluating anthropogenic changes to natural systems demand greater quantification through innovative transdisciplinary research focused on adaptation and mitigation across a wide range of thematic sciences. Southernmost Africa is a unique field laboratory to conduct such research linked to earth stewardship, with 'earth' as in our Commons. One main focus of the AEON's Earth Stewardship Science Research Institute (ESSRI) is to quantify the region's natural and cultural heritage at various scales across land and its flanking oceans, as well as its time-scales ranging from the early Phanerozoic (some 540 million years) to the evolution of the Anthropocene (changes) following the emergence of the first human-culture on the planet some 200 thousand years ago. Here we illustrate the value of this linked research through a number of examples, including: (i) geological field mapping with the aid of drone, satellite and geophysical methods, and geochemical fingerprinting; (ii) regional ground and surface water interaction studies; (iii) monitoring soil erosion, mine tailing dam stability and farming practices linked to food security and development; (iv) ecosystem services through specific biodiversity changes based on spatial logging of marine (oysters and whales) and terrestrial (termites, frogs and monkeys) animals. We find that the history of this margin is highly episodic and complex by, for example, the successful application of ambient noise and groundwater monitoring to assess human-impacted ecosystems. This is also being explored with local Khoisan representatives and rural communities through Citizen Science. Our goal is to publicly share and disseminate the scientific and cultural data, through initiatives like the Africa Alive Corridor 10: 'Homo Sapiens' that embraces storytelling along the entire southern coast. It is envisioned that this approach will begin to

develop the requisite integrated technological and societal practices that can contribute toward the needs of an ever-evolving and changing global 'village'.

**Keywords:** earth stewardship; ecosystems; Khoisan; citizen science; drone; GIS; Cape; Karoo; groundwater; ambient noise

## 1. Introduction

The biosphere is an integrated, self-regulating system that has evolved over four billion years in response to geological, climatic and biological processes. This global history can be best told across Africa [1,2]. Rapid and severe fluctuated changes in climate and habitat place living systems under extreme stress, either forcing them to adapt or face extinction. For nearly 250,000 years across southern Africa, *Homo sapiens* has been modifying the natural habitat of animal and plant species, to the point where our own survival is now threatened. As we increasingly become dependent on the natural environment to meet growing global demand for resources, a sustainable balance must be found. However, it is not clear how this may be achieved.

Cross-disciplinary studies, collectively referred to as 'earth stewardship science', are key to understanding the evolving interconnections between fauna, flora, water, soils, landscapes and our ecological role as humans. Such research requires larger multi-sourced datasets with the involvement of many more people with different backgrounds to address complex problems in an integrated manner [3].

The Africa Earth Observatory Network (AEON) is a center for Earth Stewardship Science (ESS) that provides a university-wide research and education environment to seek consilience amongst earth and life sciences, engineering, resource economics, the human sciences and politics. The network encourages cutting-edge, internationally connected, science and analytical learning using advanced tools and technologies to try solving complex problems, specifically those relating to the potential sustainability of resources in South Africa. The current aim is to impact on resources management and relate this to pressures on diversity, environment and society (poverty and well-being). AEON operates from a Commons at Nelson Mandela University, in Port Elizabeth, located along the southwest Indian Ocean (Figure 1). AEON's postgraduate researchers are from the physics, chemistry, earth and life sciences, arts, humanities and socio-political sciences, and work together with students, schools, local communities, industries, and conservation organizations. Research results are made publicly accessible through 'Mandela Talks' [4] and through design for new school curricula and education programs. The group includes stakeholders from previously marginalized groups in the data collection and interpretation, as through 'Citizen Science'. This collective approach provides opportunities to better inform local people on their impact and vulnerability across their living space, and to encourage the development of a custodial attitude towards nature. Such 'buy-in' is crucial to the success of any attempts to preserve natural and cultural heritage, especially under conditions of changing climate and population growth (Figure 1).

Southernmost Africa is a unique natural and cultural laboratory for such collaborative work. The region and surrounding ocean margins preserve an exceptional, albeit often incomplete record of more than 500 million years of plate tectonic, climate, sea-level and biological changes. This includes the best-documented and most complete history of human colonization and modification, from some of the earliest hunter-gathers close to 250,000 years ago, through the industrial and agricultural revolutions during the 18th and early 19th centuries, to present-day. The concept that human activity is changing climate and the environment is referred to as the Anthropocene, but there are many ways of defining the start of this new epoch [5]. The southern coastal region of South Africa offers a great opportunity to define this transition with accuracy because that is where humans started urbanization in numerous caves, some between 200,000 and 70,000 years ago, and today it hosts some of the world's

largest expanding cities, townships and 'favelas' (e.g., Cape Town). Using new dating methodologies, the marine fossil record and South African artefacts can possibly best define the start and evolution of the Anthropocene, which is part of AEON's transdisciplinary research.

A common goal is to explore how humans, animals and vegetation have changed, disappeared and expanded over a relatively short period of time (less than one million years), initially from a few thousand inhabitants to more than 13 million across the Cape region today. Our aim is to define and produce transdisciplinary knowledge of southern Africa's natural and cultural heritage, and package it in a way that is widely accessible, effectively producing Earth Stewards out of local communities and students.

Through the spatial analysis tools provided by Geographic Information System (GIS), integrated with the application of new algorithms, it is possible to quantify the development and evolution of the South African landscapes and the human role, with acceleration in the modern era. This work feeds data to larger AEON's programs like 'Iphakade: Observe the present and consider the past to ponder the future' [6].

Recent episodes of severe drought and flooding in South Africa are having increasingly important repercussions for groundwater sustainability planning and soils stability that form the basis of food security, as well as preservation and development of infrastructures. Here, we demonstrate how this region can be used as an earth-humanities baseline against which to measure and predict such changes into the future.

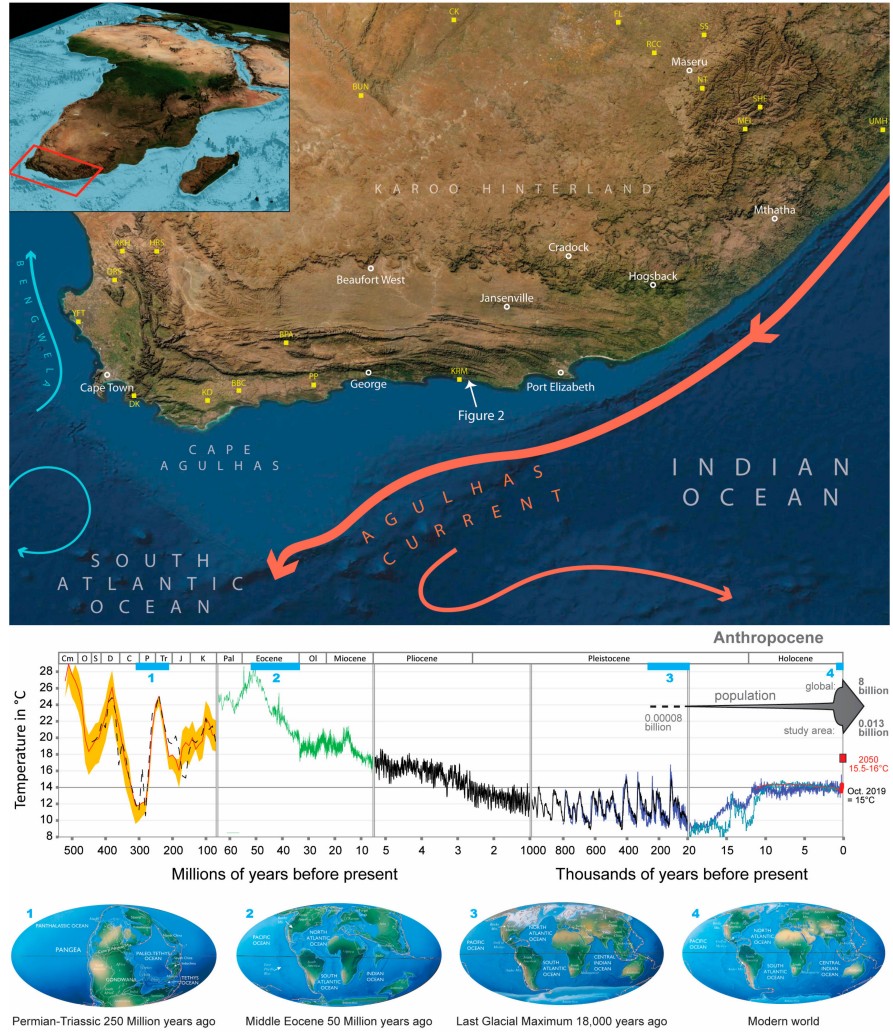

**Figure 1.** Map of southernmost Africa (inset for location), a unique 'Cape-Karoo' region flanking the

Map of southernmost Africa (inset for location), a unique 'Cape-Karoo' region flanking the South Atlantic and Indian oceans, which demands a transdisciplinary approach to understand the interactions between geological, biological and cultural processes; yellow squares are well-known archeological heritage sites. The graphical curves below show variations in surface air temperature at different time-scales, from 540 million years ago (Ma) to Present [7]; red box shows the range of temperature estimates by 2050; and grey arrow illustrates the rapidly expanding population (grey numbers) of *Homo sapiens*, between 250,000 and 70,000 years ago, and from the local Khoisan (Nau-Omkhai ku Dara) to modern colonization over the last 400 years. It is at the start or during this period that different definitions of the Anthropocene are best recorded. Paleo-globes [8] at the bottom illustrate the geodynamic and climatic settings during key periods of global changes, the records of which are exceptionally well preserved across this region, including across Table Mountain flanking Cape Town.

## 2. Natural and Cultural Heritage Changes across Southernmost Africa—What is at Stake?

To aid the design of long-term monitoring and conservation strategies for the studied region, AEON is compiling a scientific database of significant natural and cultural environments. This public information will be essential to preserve its linked Cape coastal region and Karoo hinterland services in the face of anticipated climate changes.

### 2.1. Natural Heritage: A Buffer Zone

The origin of the southern tip of Africa links to the formation of the Gondwana supercontinent and the subsequent separation of Antarctica and South America from Africa some 180 Ma during the emergence of the southern oceans [9]. Geodynamically, it is a buffer zone between the continents and the oceans. With the successive opening of the Indian and South Atlantic oceans, long-term global cooling began, sea and land levels rose, and new circulation regimes were established around the southern Cape (Figure 1). These changes led to periodic glaciations at the Polar Regions, which, in turn, increased variations and frequency in precipitation and sea-level changes. These fluctuations in the environment have largely driven the evolution of the present ecosystems, including the first humans along the coast.

Interactions between the cold Benguela current that upwells from deep regions of the southern Atlantic to the west coast of southern Africa, and the warm Agulhas current flowing from the equator southward along the east coast provide unique, highly diverse environmental conditions for the region's marine and terrestrial life. Nine marine bioregions are recognized, of which the southern Cape (Agulhas), with its high level of endemic marine fauna, predominates [10]. On the adjacent continent, all the major South African biomes occur, including Afromontane mist-belt forest and grassland, coastal forest, Cape Fynbos, and Albany thicket that all link to the dry (Karoo) internal savannah [11].

Landscape archeology that is based on evidence of early human habitation is unique across the studied region (Figures 1 and 2). Here, the earliest evidence of our cultural evolution have been re-discovered, including: Acheulean stone tools (e.g., Montagu Cave, 500–200 ka); the oldest known engraved rock and some necklace beads made from gastropod shells (Blombos Cave, 75 ka); a variety of ostrich eggshell containers (Diepkloof Rock Shelter, 65–55 ka) [12,13]; and the earliest human footprints along sand dunes (Langebaan, 117 ka) [14]. These, together with the dated sedimentary records, reveal strong relationships between human evolution, sea-level and climate changes, with the most important modifications occurring during intervals of maximum glaciation ('ice ages'), such as the Cape coast is envisaged to represent the cradle of human culture.

Around Cape Town, there are more tragic archives of the recent stories of how the first European colonizers removed, killed and incorporated the indigenous San hunter-gatherers and the Khoi pastoralists as slaves [15]. This knowledge has, for example, recently motivated the Khoisan to advocate the reburial of Sschura (Sarah Baartman), in 2001 on Vergaderingskop in the Eastern Cape. This 'First Nation' is re-experiencing their spiritual connectedness to the past, and which compels

current chiefs and activists to seek restorative justice for the loss of identity, tradition and land, based on archeology and landscapes that can now be carefully re-mapped using drone technology (Figure 2).

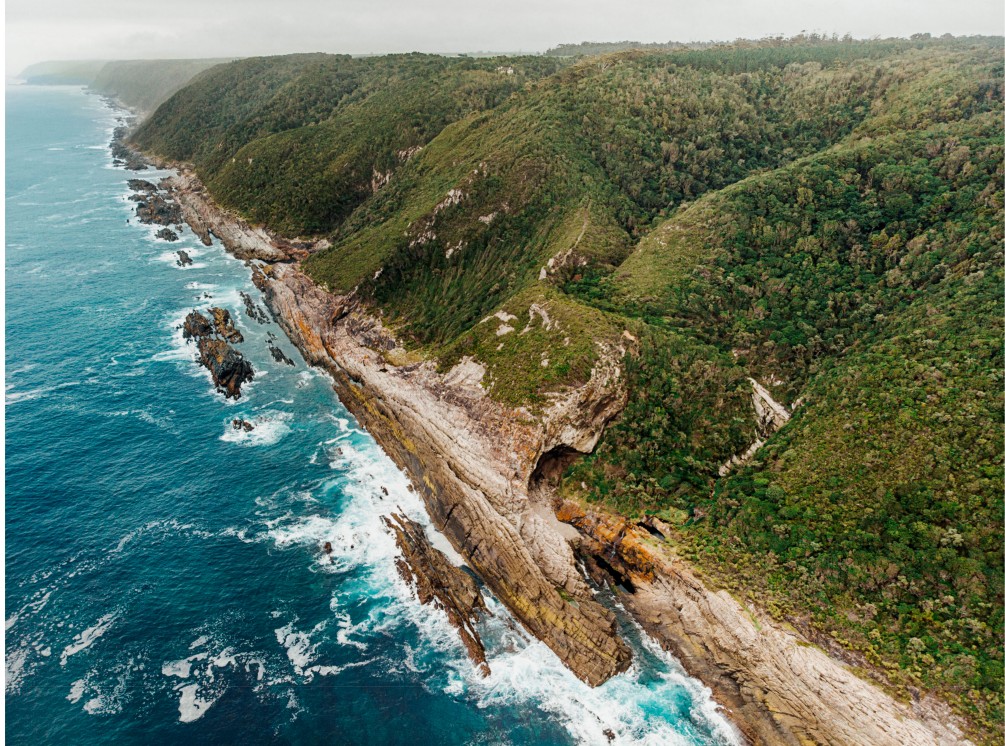

**Figure 2.** Aerial drone photo of a cave eroded in synclinal folds of the Table Mountain Group quartzites, bordering the Tsitsikamma indigenous forest along the southern coast (Figure 1 for location; see http://aeon.org.za/remotesensing/ for drone video). The flat erosion surface elevated at more than 200 m above sea level is a wave-cut platform that records southern Africa uplift and fluctuated global sea-level drop since about 100 Ma.

*2.2. Cultural Heritage: 'First Come-Last Served'*

The Khoe|Xam (Khoi and San) people were the first people to live and thrive along this coastline [15,16]. Since the colonial invasions began in the 17th century, South African indigenous cultures have faced intense discrimination, leading to deculturation and near extermination. Today there is a minimal reference in education to indigenous knowledge that reflects the depth of the ancestral opinions that influenced African thinking [17,18]. The Khoe|Xam are trying to re-identify and to conserve the integrity of their cultural knowledge by teaching the youth about their heritage. Today their geographical roots, and sense of belonging, entitlement and ownership, legitimizing their ethnonational grouping still needs considerable scientific observation and quantitative data.

Through our Khoisan research unit at AEON (!Nau-Omkhai ku Dara), we use two approaches in knowledge generation, development and utilization to learn and preserve Indigenous Knowledge (IK) for future generations.

- First, cultural mapping is used for identifying and analyzing the cultural assets of a community. These assets can be viewed as tangible (e.g., rock art) and intangible (sense of place–which includes beliefs, memories, norms and values; e.g., [19]).
- Secondly, we are using narrative inquiry to systematically gather, analyze and represent the Khoisan stories as told by them. These narratives encapsulate personal and human elements of Khoisan lives over time, taking into account the connection between individual experience and the cultural context [20].

AEON's transdisciplinary research helps for dialogue, engagement and co-determination around the narratives of the past, present and future of the first indigenous people [21]. This follows the early fieldwork of geologist George Stow who examined rock paintings and recorded the stories and visions of the Khoe|Xam [22]. These texts form a unique archive of the Khoisan, whose language and culture have been mostly extinguished [23,24]. One of the challenges now is to use drone technology to decipher links between their past living environments (caves and shelters), ecosystem services and remaining stories of the Khoisan chiefs (Figures 2 and 3).

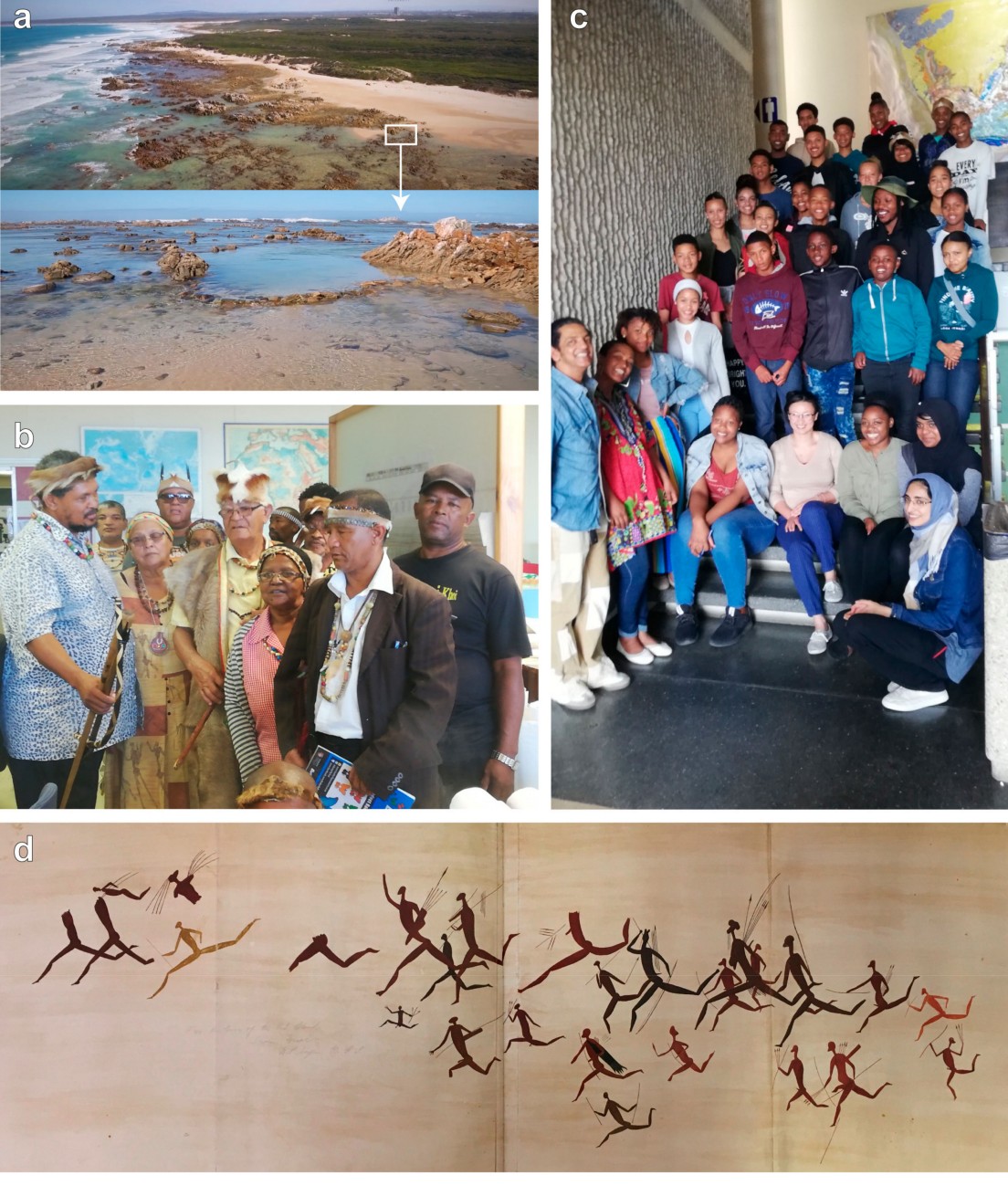

**Figure 3.** Indigenous Knowledge (IK) documentation through Koi-San research at !Nau-Omkhai ku Dara: (**a**) Fish-traps at Cape Recife nature reserve, some of the earliest human changes to the ecosystem along the coastline (drone data across these traps is shown at "Mandela Talks' [4]); (**b**) Khoisan chiefs debating their socio-cultural challenges in AEON's Commons; (**c**) youth engaging in discussions concerning their future career with AEON's masters and doctorate graduates at Nelson Mandela University; and (**d**) rock painting, copied and first published by geologist George Stow [16].

### 3. Linking Geology and Biology through Hydrocensus—Clean Water for All

The supply and distribution of potable water are one of the major challenges currently facing the communities across the studied region, as illustrated by the onset of the 2015 drought in Cape Town. By the end of the dry season in May 2017, the drought was declared the city's worst in a century [25–27]. Then in February 2018 and October 2019, many other towns across the region were declared drought disaster areas following the several years of precipitation well below average rainfall; e.g., Port Elizabeth received 42% below the average amount of 271 mm between January and July in 2018 [28]. This has emphasized that links to groundwater are vitally important to South African communities, agriculture and livestock, in particular, inland across the semi-arid Karoo (Figure 1 for location).

In anticipation of the pending water crisis, we are conducting surface and subsurface (ground) water surveys, i.e., hydrocensus. These baseline studies use and complement the geological, geophysical and biological mapping, as highlighted below.

*3.1. Geological and Geophysical Mapping*

Field geology and geophysics contribute to a holistic understanding of the evolution of the South African margin. Surface and subsurface data gathering provides the backgrounds needed to produce the detailed maps of our natural and cultural heritage.

- At large-scale (>10 km) Landsat8 Operational Land Imager data, processed using Microimage TNTmips software, assist in mapping different crustal domains, such as the Pan African (Gondwana) suture zones and the Cape Fold Belt along the coastal region (Figure 4a). The two short-wave-infrared bands (7 and 6) and one near-infrared band (3) are combined to create RGB images that provide additional lithological and structural information that is not always evident in the field.

- At outcrop scale, we make use of drone technologies to create ortho-photomosaics of the ground and vertical cliffs (<500 m$^2$) in Agisoft Metashape software. This photogrammetry method requires processing large point clouds and several iterations to produce base-maps and digital surface models at a resolution of under a meter (about 5 cm per pixel for a flight height of 30 m). These 3D models are very useful for delineating geological contacts, structures (dykes, joints, hydrothermal vents) and help to compute drainage systems.

The construction of geological maps requires extended periods of fieldwork, often camping, while exploring remote locations, thereby making essential engagement with the local community, farmers and traditional leaders (chiefs). For example, our work in the Drakensberg and Lesotho Mountains has led to the recent discovery of pillow lava structures, which attests to a marine environment linked to the opening of the Indian Ocean some 180 Ma [29]. These rock formations are preserved at present-day elevations of more than 1700 m, making it difficult to fit into current plate tectonic models and sea-level changes. More fieldwork with many students is needed to map and accurately date the evolution of this paleo-coastline that holds important implications for our understanding of the geomorphology and uplift history (epeirogeny) of the studied region.

To further constrain the geodynamic evolution of the South African margin, our models integrate offshore seismic and borehole data (Figure 4b). On the continent, the sedimentary sequences are in part eroded and often link to shallow water facies, with few fossils preserved, and thus, are difficult to date and correlate over large areas. The fieldwork is, therefore, associated with the integration of offshore geophysical datasets with important spatial coverage (available from the Petroleum Agency of South Africa), which help better defining the different stratigraphic units and reconstruct the sedimentation history from source to sink [30,31]. The seismic interpretations are then integrated into Kingdom Suite and ArcGIS to construct isochron and isopach maps that form the baseline input data for the plate tectonic and paleo-environmental reconstructions to constrain vertical motion of the continent and sea level. Across the studied margin, we have now identified several shallow to intermediate depth

stratigraphic units from the Kimmeridgian (~155 Ma) to present-day time. This new framework reveals distinctively different tectonic styles and basin dynamics, first during the separation between Africa and Antarctica (180 Ma onward), and then during the subsequent movement of South America relative to southern Africa (starting 135 Ma).

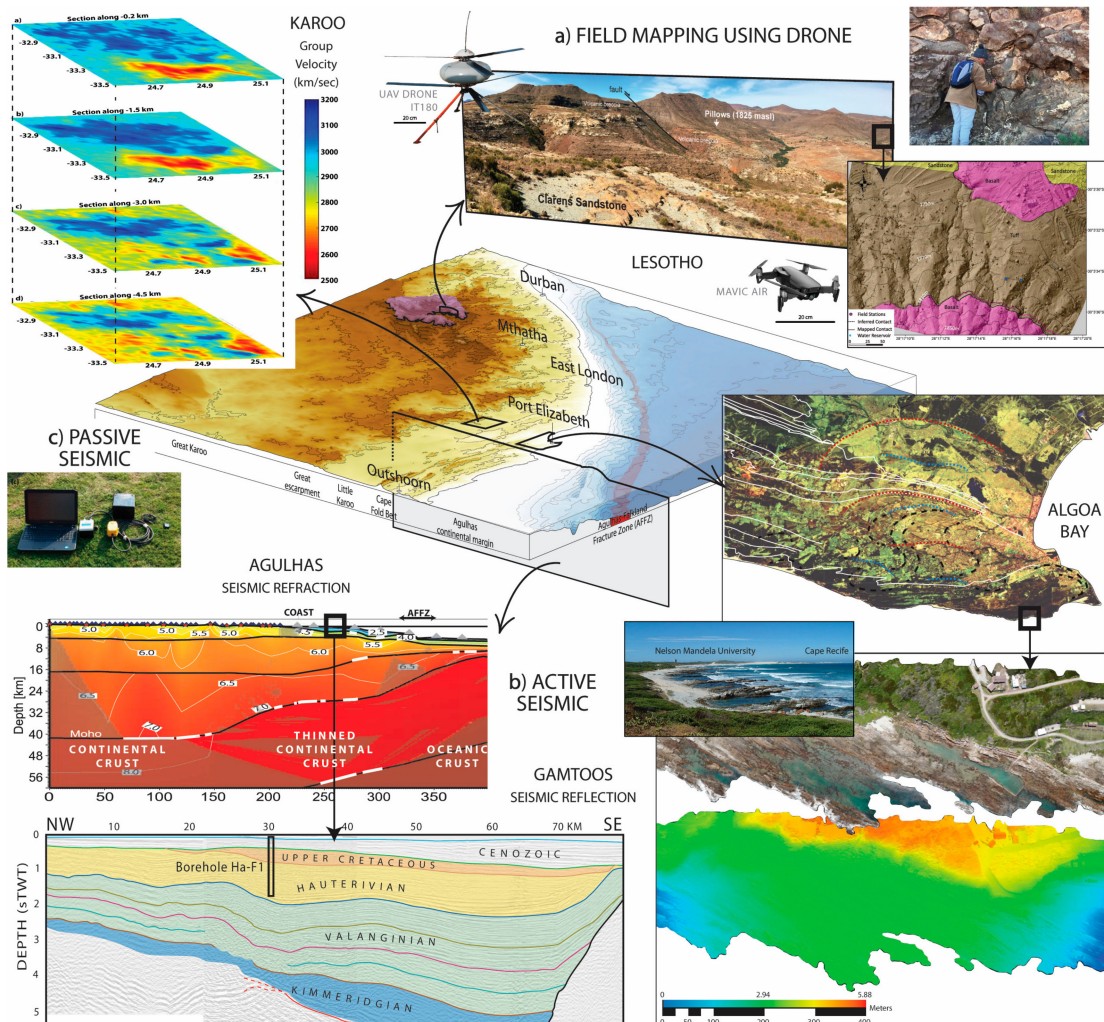

**Figure 4.** Geological and geophysical data collecting methods across the south-eastern corner of the South Africa margin (ETOPO 2 min precision elevation model), from the Drakensberg Mountains, ca. 3500 m above sea level (asl), to the abyssal plain of the southwest Indian Ocean, ca. 5000 m below sea level (bsl): (**a**) Mapping the paleo-coastline in Lesotho and the modern rocky shore near Port Elizabeth using satellite, drone, and field surveys; (**b**) interpreted seismic refraction and reflection profiles imaging the structure of the crust onshore and offshore; and (**c**) passive seismic tomography in the Karoo hinterland, based on 'sound of waves' from the ocean (see text for details).

Our ongoing seismic experiments have also shown that ambient seismic noise, generated from pounding of ocean waves that surge the coast, can be used to map the subsurface virtually anywhere inland (Figure 4c). This relatively new technique of tomography is based on cross-correlation of random diffuse ambient seismic signals, recorded between pairs of seismic stations, to approximate empirical Green's functions [32]. Green's function between two stations can be seen as a seismogram, recorded at one station from a source originating from another station. Prior to pre-processing, the continuous ambient seismic noise waveform data from the triaxial components are converted from Cube to Miniseed format and divided into daily (24 hours) segments. Additional format conversion for waveform

analysis is done in Matlab, Msnoise and Msnoise-TOMO. Pre-processing is automated and completed for each station to prepare the waveforms for correlation [33]. After cross-correlation between all station pair's, dispersion measurements and tomographic inversion are performed. The local dispersion measurements are jointly inverted to create isotropic shear velocity (vs) models down to more than 5 km in depth, where the major geological structures can be interpreted. This technique is now being further developed for imaging shallower processes, such as for the monitoring of dam stability (see 4.3) and surveying animal behavior. These processes need to be dated using geochronology techniques using, for example, U-Pb and Rb-Sr isotopes, and sclerochronology, as described below.

### 3.2. Listening-to and Timing-Off Evolving Marine Ecosystems

Geophony (sound from geological processes), biophony (sound from biological sources), and anthrophony (sound produced by humans) create unique soundscapes essential to sound-dependent species for orientation and navigation, as well as for communication with conspecifics. Global ship density has increased over the last decade, thereby increasing anthropogenic noise level and altering soundscapes [34–37]. Shipping noise may change marine species behavior or distribution pattern within affected habitats [38–41]. Acoustic monitoring of underwater soundscapes in conjunction with observations on animal behavior and distribution patterns is helping us to assess the effect of shipping noise on large baleen whales, which are low frequency hearing specialists, and therefore, particularly sensitive to low frequency vessel noise.

We monitored the soundscape in St. Francis and Algoa Bay (Figure 5a) with three autonomous SM2M+ acoustic recorders equipped with an HTI-96-min hydrophone (sensitivity -166 dB re v/μPa), from January 2015 to December 2017. In addition, the automatic identification system (AIS) data for all ships traversing the bay in 2015 was obtained From FleetMon [42]. Data were processed in Matlab with custom written codes to test spatial differences in ship noise. We are then using spatial analysis in ArcGIS to assess whether humpback whale (*Megaptera novaeangliae*; Figure 5b) and southern right whale (*Eubalaena australis*; Figure 5c) mother-calf pair distribution patterns are affected by received levels of ship noise. This serves as a baseline to monitor for marine animal behavioral changes under future port development plans that are likely to further increase ship noise level within Algoa Bay.

Sclerochronology [43,44], which is the study of periodicities preserved in growth patterns of layered biogenic carbonates, such as stromatolites and bivalve shells (Figure 5d,e), is also an ideal quantitative tool to reconstruct the environmental changes across Algoa Bay. Using geochemistry, we can translate the growth and temporal record of the organism into relative proxies for physical parameters, such as temperature, seasonality, tides and salinity.

With the aid of the University of California Los Angeles (UCLA), we test carbonate clumped isotope thermometry to directly measure the temperatures, both from modern organisms and their fossil equivalents, including gastropods, mollusks, corals and shark teeth (Figure 5f). This method has been applied successfully to stromatolites found in intertidal pools, near Port Elizabeth, where the water temperature is monitored. This establishes a new geochemical baseline for the studied region, which was only possible with several visits at UCLA and online discussions as part of the ESWB-Environmental Science Without Border program [45].

In the older and more complex oyster shells and shark teeth samples, carbonate layers have distinctively different compositions [46], which must be mapped at higher resolution under the microscope for a more precise sampling of the different phases (e.g., chalky vs. fibrous calcite). In addition, in the fossil samples, the extent of alteration and diagenesis must be determined. This is being achieved using high resolution Scanning Electron Microscopy (SEM) and Raman spectroscopy at Nelson Mandela University, where diagenetic alteration and the occurrence of recrystallized minerals can be quantified.

We aim to link modern and past environmental conditions along the coast, with analytical techniques that require intensive laboratory work and new micro-drilling tools. In addition, a remaining

major challenge is to precisely date the fossils (at an accuracy of thousands of years over a period of some 6 million years) to calculate rates of climate changes and the onset of the Anthropocene.

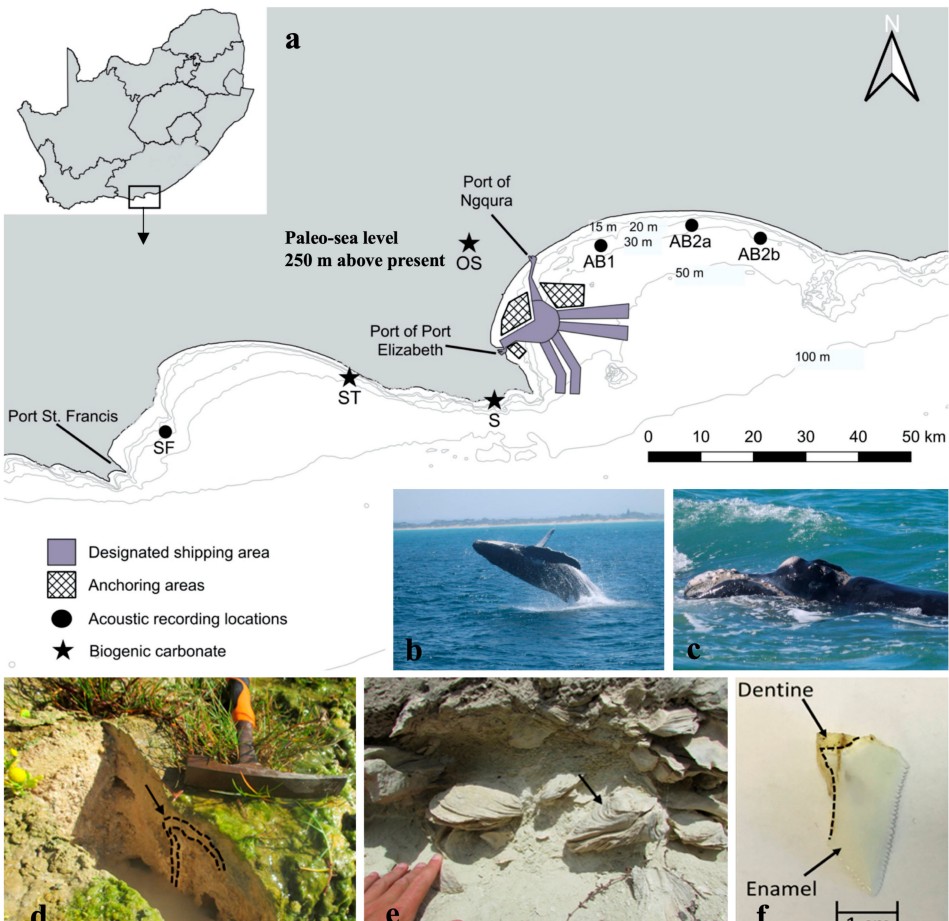

**Figure 5.** Soundscape ecology and geochemical studies of St. Francis and Algoa Bay, near Port Elizabeth: (**a**) Map showing the location of acoustic loggers and biogenic carbonate samples; (**b**) humpback whale (*Megaptera novaeangliae*) calf; (**c**) southern right whale (*Eubalaena australis*) calf; (**d**) stromatolites (site S); (**e**) Plio-Pleistocene oyster fossils (*Striostrea margaritacea*; site OS); and (**f**) modern shark teeth (*Carcharodo carcharias*; site ST).

### 3.3. Mapping Ecosystems on Land Using Specific Patterns of Animal Behavior

GIS and drone technologies are also used to study the ranging behavior of animal populations, habitat use and environmental interactions, group sizes and distributions, migrations, and minimal vegetation requirements for different species. For example, termites, through their interactions with the surface and subsurface, act as reliable indicator species to track natural and anthropogenic disturbance effects across the top tens of meters of soil. We are analyzing the spatial patterns of termite mounds using satellite and aerial images, drone digital surface models and ground surveys (Figure 6). Results show that drone imagery captures a larger number of mounds than can be mapped through simple aerial photographs, and that the mean nearest neighbor distance between termite mounds determined by field measurements can be used interchangeably with that computerized from the drone ortho-mosaics. Any changes in these geometries over time will, therefore, be able to be linked to a specific ecosystem and climate changes.

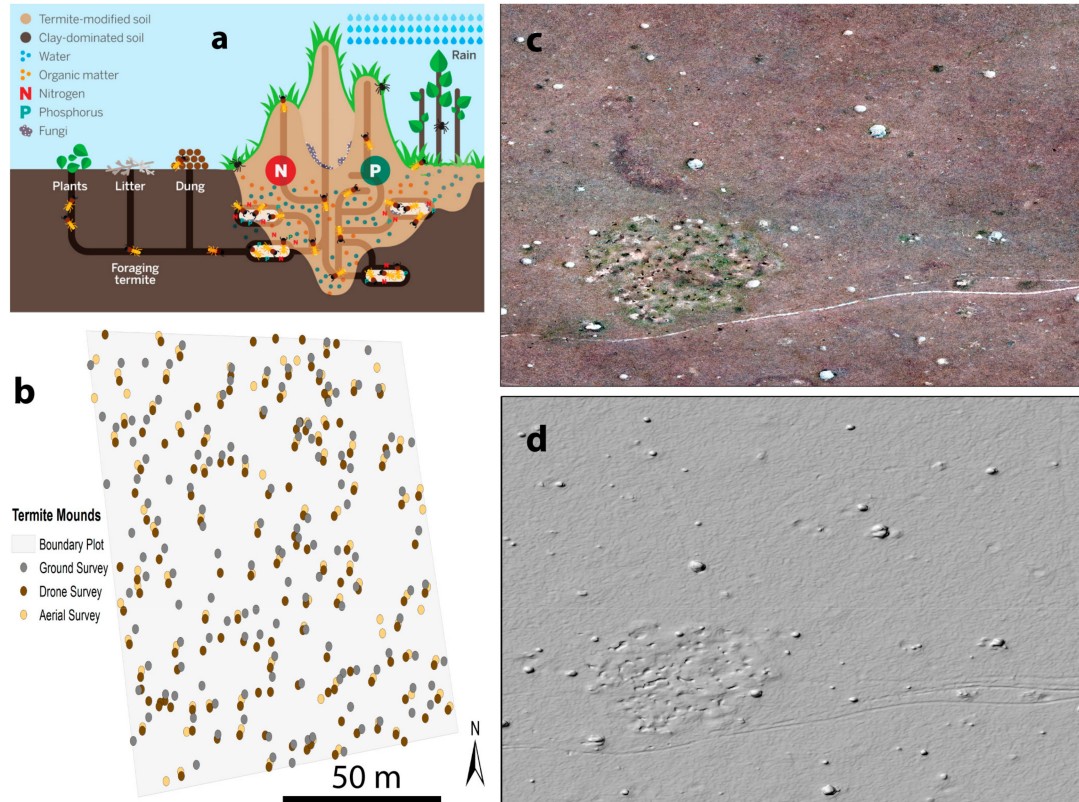

**Figure 6.** Spatial distribution of termite mounds across a 2,000 m² surveyed area of the Karoo, near Cradock (Figure 1 for location): (**a**) Architecture of a termite mounds [47]; (**b**) map using ground, drone and satellite image surveys; (**c**) ortho-mosaic; and (**d**) digital surface model.

The biological richness and high endemism of montane regions illustrate the impact of geological features on biological diversity. This diversity is created by micro-climatic conditions of escarpments and the island-like isolation of high altitude habitats. Consequently, mountains often constitute biological hotspots and priority areas for conservationists. The Amatole Mountains are located along the Karoo escarpments, approximately 100 km inland from the coast. In this area, near Hogsback (Figure 1 for location), there are good examples of this biogeographic principle, making it a fascinating natural laboratory for studying the evolution of plants and animals.

Endemic animals in this area are characterised by small body size and low mobility. This is the case, for instance, for the legless skink (*Acontias breviceps*), two frogs (the Hogsback chirping frog *Anhydrophryne rattrayi* and the Hogsback caco *Cacosternum thorini*), and a small toad (the Amatole toad *Vandijkophrynus amatolicus* [48]). The recent discovery of the Hogsback caco (*Cacosternum thorini*) shows how the local geology influences species diversity and offers an opportunity for collaborative research involving remote sensing, and specifically for drone data capture. These tiny (15 mm) frogs, identifiable by their unique, diagnostic call, are found in a very small area around Tor Doone [49]. They inhabit the very rare and localised habitat of deep oligotrophic pools in high altitude peatland. In this case, it is found immediately behind a 20 m thick, subvertical Karoo dolerite dyke that forms a natural barrier in the drainage system (Figure 7a), allowing the pools to remain full for several weeks and thereby creating the preferred habitat of the frogs (Figure 7b). Because each pool is defended by a single male (Figure 7c), drone surveys are being used to count the pools in order to estimate the population size and their changes during climate variability.

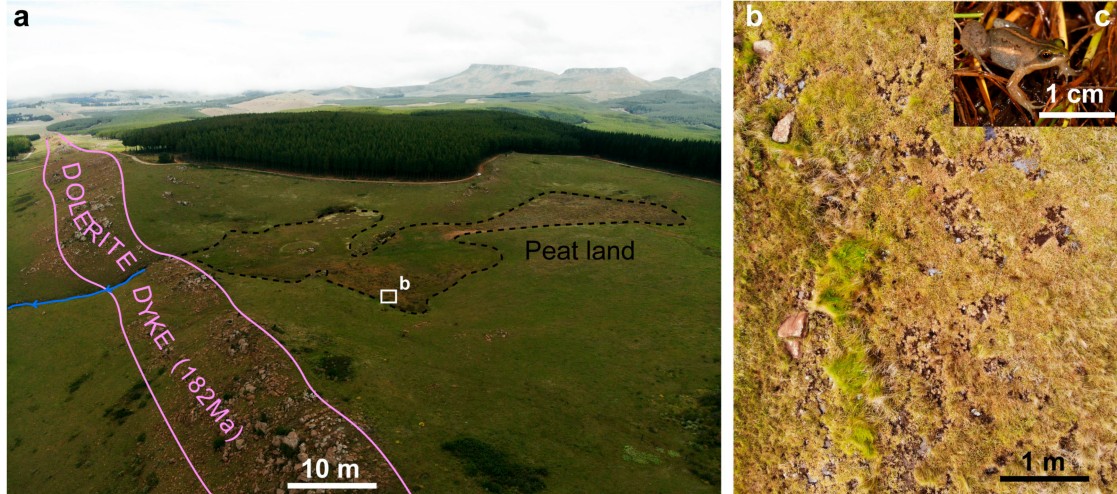

**Figure 7.** Biological survey by drone at Tor Doone, near Hogsback (Figure 1 for location): (**a**) Aerial photo of the peatland habitat upstream a dolerite dyke intruded at 182 Ma; (**b**) water pools decimeters in size and depth; and (**c**) *Cacosternum thorini*, one of the rarest frogs in South Africa (photo by Werner Conradie).

In dry regions like the Karoo, mountains are also attractive to people, creating conflicts between landowners and wildlife. This is well-illustrated by the field base of our APIES group (African Primate Initiative for Ecology and Speciation) in the area around Hogsback.

- A primary, passive conflict and major threat is habitat destruction caused mainly by the forestry industry. The natural forest has virtually been extirpated on most of the surrounding mountains in favour of the pine plantations that drive local forestry industries in that indigenous plants only subsists on the steeper slopes of the escarpments.
- The second, direct conflict is limited to targeted species considered "pests", taking the form of trapping and shooting. In Hogsback, the targets of such persecution are those monkeys that raid gardens and orchards, primarily the chacma baboons (*Papio ursinus*), but also the endangered and protected Afromontane samango monkey subspecies (*Cercopithecus albogularis labiatus*). Studies of human-wildlife conflict are fundamentally interdisciplinary and involve both social and natural scientists.

Our case study, started in 2010, centres on two main methods, one subjective: Questionnaires that have revealed the common perception that the village has been invaded by baboons and monkeys. The second approach is objective: Regular counts of animals and following the populations over time has revealed higher population densities in the village (although there are much more limited raiding of residential properties than the questionnaires imply), and very stable troop and population sizes over time (Figure 8a). Another complementary study, initiated in 2018, investigates the role of the monkeys in forest regeneration by seed dispersal, and the possible inverse role of seed predation in the case of invasive plants; two ecological services that potentially make these animals very useful in maintaining the relict indigenous forest (Figure 8b). All these field studies involve remote sensing and mapping in ArcGIS to compare the distribution of the animals with those of their resources, including human resources (Figure 8c). Our long-term objective is to contribute to the conservation of the indigenous mist-belt forest and its unique fauna and flora (Figure 8d).

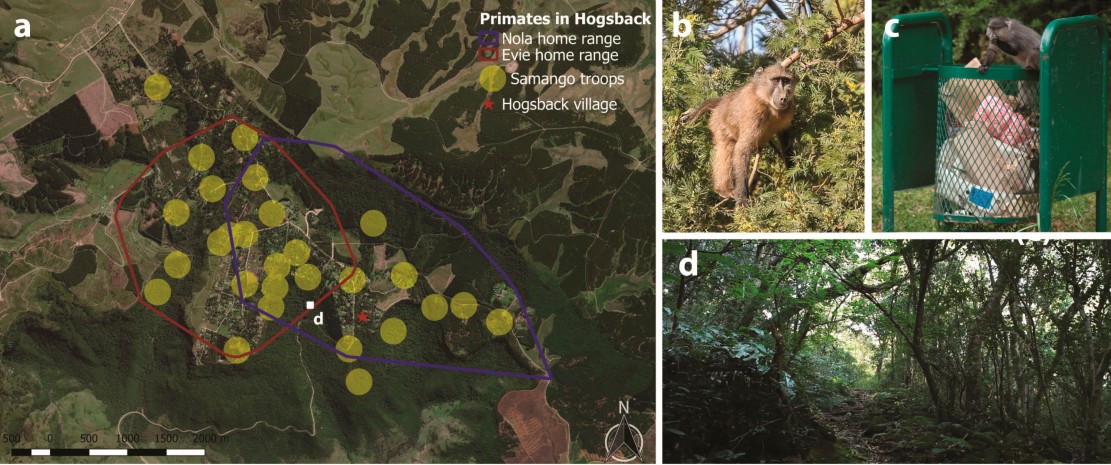

**Figure 8.** Baboon and samango troops living in Hogsback: (**a**) Distribution map; each yellow circle represents a samango troop core area; red and blue polygons represent the maximum home range extension for the two baboon troops studied over 10 years. (**b**) Juvenile baboon on black wattle tree (*Acacia mearnsii*); (**c**) samango monkey eating from an uncovered trash bin; and (**d**) indigenous forest.

Remote sensing has become an essential component of the panoply of techniques used for conservation and wildlife management. In particular, camera trapping is commonly used to survey rare and elusive animals, testifying to a visually-oriented view of natural landscapes and anthropogenic changes. More recently, we have started to apply a similar technique using sounds, defining the new field of soundscape ecology [50].

Sound-trapping is an important complement to camera trapping that allow long-range detection of animals. An interesting application of the method is to help resolve the controversy around the number of elephants still present in the studied area, because elephants communicate over long distances using infrasonic vocalisations and percussion. If several individuals are present, infrasound detection, and vibrations (seismic waves) should allow us to detect their presence. Sound-trapping can also be used to test the influence of soundscapes on evolution, and particularly speciation in vocal animals like the frogs and primates.

### 3.4. Hydrocensus

As part of the baseline studies, we focus on measuring and analyzing the surface and groundwater resources, particularly in the semi-arid Karoo. With this hydrocensus we aim to better understand the natural and anthropogenic components of the hydrological cycle. This will allow for informed decision-making and progressive legislation in near future.

A key aspect of hydrocensus is studying the different sampling methods that are used to collect the primary data and water for analysis. Multiple tools are tested to increase reliability that include: Temperature, pH, eH, static water level, and electrical conductivity loggers and probes. The groundwater from equipped boreholes is collected from an outlet and sampled at depth using a bailer, discrete-interval sampler or a low flow pump. The collected samples are then kept in high-density polyethylene (HDPE) bottles to avoid contamination and refrigerated prior to transport to the laboratory.

Two testing sites include: A natural wetland at Rietvlei, near Cape Town [51], and a small drainage basin where historic uranium (U) rich mine tailings are exposed at Rietkuil, near Beaufort West (Figure 9a). U concentrations are measured from the different surface water bodies and the shallow aquifer (Figure 9b). Initial results reveal that U concentration in surface water samples (2.50 µg/L on average) is significantly lower compared to that in the groundwater (13.60 µg/L on average). This is lower than the permissible limit of 30 µg/L U recommended for safe drinking water [52,53]; however, a risk remains as groundwater is primarily used for domestic and livestock. GIS spatial analysis is

further used to map the distribution of recharge rates of the groundwater table, which is, in turn, influenced by the type of soil and relief (Figure 9c).

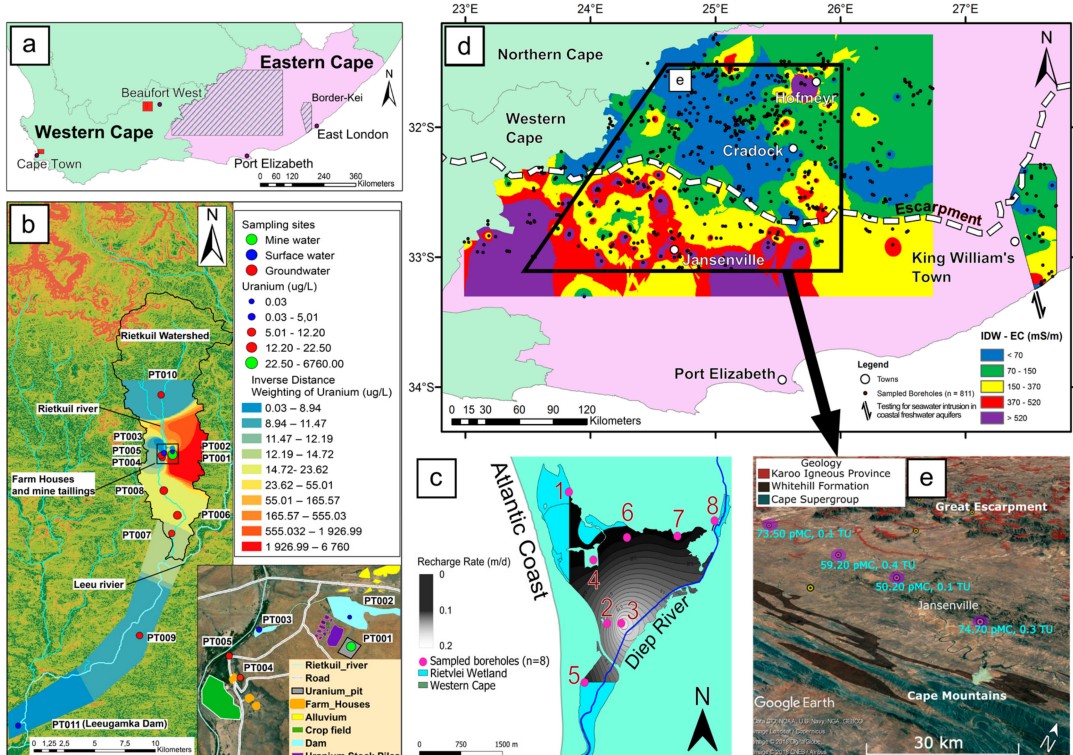

**Figure 9.** Mapping surface and ground waters of southern South Africa: (**a**) Location map of these baseline studies (hatched areas) and the two testing sites (red boxes); (**b**) measured Uranium (U) concentration levels along the Rietkuil drainage near Beaufort West, with a satellite image showing the mined pit, stockpiles, and farmhouses; (**c**) groundwater recharge map of a wetland at Rietvlei, near Cape Town; (**d**) interpolated Electrical Conductivity (EC) measurements of groundwater in the southern Karoo; and (**e**) isotopes signatures in groundwater samples (purple circles) near Jansenville that show mixing of old/deep saline waters with recent/shallow freshwaters (pCM = Percent Modern Carbon; TU = Tritium Units).

Across larger areas of the Eastern Cape, groundwater studies were conducted to identify sensitive areas for long-term monitoring, between Cradock and Jansenville, and along the coast, near King William's Town (Figure 9d). In both regions, inverse distance weighting has been used to map the groundwater electrical conductivity (EC). This clearly shows higher salinities south of the main escarpment and near the coast. In these areas, possible sources are further determined using isotope analysis [54–56]. It was initially proposed that the relatively higher salinity south of the escarpments may be due to low precipitation and limited dilution of groundwater with fresh meteoric water [57]. However, new groundwater samples collected in this area have both 14C (Carbon) and 3H (Tritium) isotope signatures of deep (old) groundwater, usually saline and non-potable, mixed with shallow, modern groundwater, with ca. 50–74 Percent Modern Carbon (pMC) for 14C (Figure 9e). This suggests that pathways, such as natural fault systems exist, allowing groundwater from depths of 2–5 km below the surface to migrate to shallow depths where it cools. By contrast, using 2H and 18O stable isotopes, the gradient of salinity measured along the coastal region is proposed to be derived from saline marine aerosols concentrated on the soil surface and adsorbed into groundwater during precipitation or irrigation.

## 4. Monitoring Rates of Environmental Changes

Monitoring our physical environs is critical to better understanding dynamic natural and anthropogenic processes, which aid in developing strategies towards future sustainable management of key resources, such as food, water, land and infrastructures under accelerated population growth as the global population is predicted to surge to an estimated 9.7 billion by 2050 [58]. The growth will place intense pressure on already strained natural resources required to support the ever-increasing demand for food, feed, fuel and water in a time in which global economies are bound to sustainable development and the implementation of socio-economic welfare. To meet these demands, the future sustainable utilization of land so it is able to support vital services, products, and infrastructure become vital [59,60].

Understanding the geomorphology of the Karoo hinterland, dominated by elevated plateaus of igneous and sedimentary rocks, requires a different analysis from that of the Cape coastal areas where rivers mostly cut through vertical, hard quartzites [61]. Recent work has shown that many Karoo surfaces have slow rates of denudation of fewer than 4 m/Ma, which suggest low soil production, and therefore, a high risk of unsustainable agricultural practices [62,63]. Moreover, there are regions where high erosion rates affect local villages.

### 4.1. Soil Erosion

Erosive gullying is a prominent form of land degradation during the Anthropocene (Figure 10a). With more than 150,000 ha of land currently exhibiting erosional gullies, the Eastern Cape is classified as having a moderate to high erosion risk driven by both natural and anthropogenic factors [64]. The presence of quaternary colluvium and pedogenic sediments of the Masotcheni Formation mantling low gradient hillslopes near Mthatha make this area particularly susceptible to widespread erosive gullying (Figure 10b). Gullying is here exacerbated by the region's history of inappropriate land use practices and population growth. These erosional gullies pose a hazard to existing and planned infrastructure developments in an area seen as one of the fastest-growing towns in the region.

Detailed multi-temporal mapping of the gullies from GoogleEarth imagery, historical aerial photographs and drone surveys, is used to document the expansion and growth patterns of these erosional features over time. Spatial analysis into the evolution of one of the larger erosional features observed just 6 km north of Mthatha (the Ngwevana Gully) revealed an aerial land expansion of the gully from 8968 $m^2$ (0.8968 ha) to 34,680 $m^2$ (3.468 ha), which translate to a 386% growth between 1938 and 2019 (Figure 10c–h). This shows an average expansion rate of 0.58 m/year, with a faster average head ward erosion rate (0.73 m/year) in comparison to its average lateral expansion rate (0.43 m/year). Assessing such growth rates over time will aid in land use planning, conservation and management of many areas on which the livelihoods of local communities depend.

### 4.2. Conventional or Organic Farming

One-third of the world's land has been degraded, and predictions show that food production needs to double by 2050 to keep pace with the world's growing population [65]. To achieve these targets farming practices, particular across Africa, must be conducted in a sustainable way and limit further damage to our natural resources. Agricultural activities are currently responsible for 30% of greenhouse gases in the atmosphere. Between 1960 and 2008 food production increased by 150% [66], with fertilizer usage increasing by 800%. Population and economic growth are some of the most important drivers of CO2 emissions and thereby influencing climate change.

In 2014, long-term comparative farming systems research trials were set up at Nelson Mandela University (George Campus). We are comparing their conventional (con) and organic (org) farming methods at four sampling sites (Figure 11a). The two different farming methods are assessed for CO2 emissions and relevance to climate change for sustainable food production. A Picarro G2201-I gas analyser was used to analyse the emitted gas trapped in polypropylene cans placed in every sampled

plot (Figure 11b). Mean CO2 emissions from the conventional soils (R2 T6 con= 64.887 ppm and R1 T6 con = 66.424 ppm) are higher than those measured from the organic soils (R1 T3 org = 53.264 ppm and R2 T3 org = 47.885 ppm). Results also show that high moisture content likely inhibits microbial activity resulting in decreased CO2 emissions (Figure 11c). It is, therefore, important to minimise greenhouse gas emissions from the soil during the process.

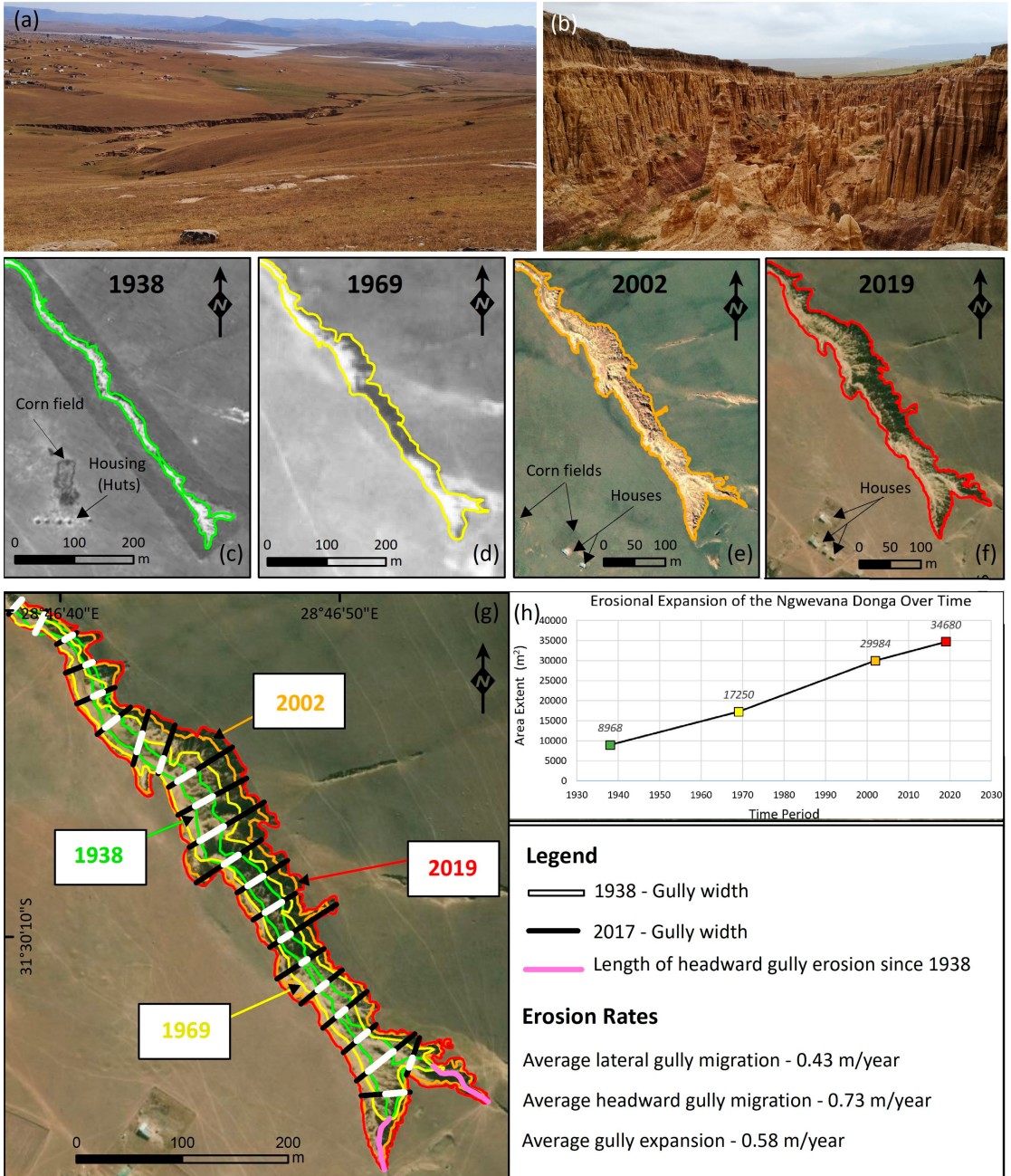

**Figure 10.** Spatial mapping of erosion rates at the Ngwevana Gully, near Mthatha (Figures 1 and 4 for location): (**a**) Landscape view looking northwest; (**b**) outcropping Masotcheni Formation and Beaufort Group rocks into which the gully is incised; (**c–f**) individual aerial views outlining the perimeter of the gully at different times; (**g**) superimposed gully perimeter outlines showing consistent lateral and head ward gully expansion; and (**h**) plotted results show an overall increase in gully area from 8968 m$^2$ to 34,680 m$^2$ over the last 81 years.

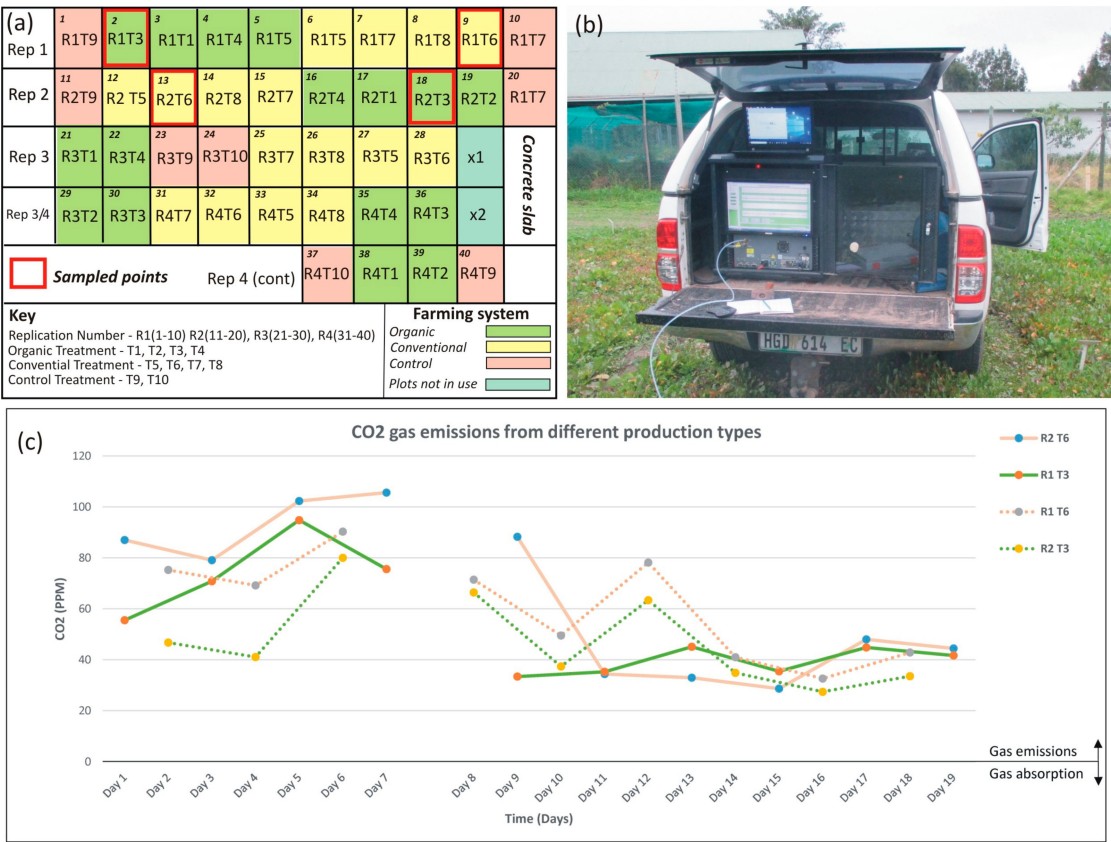

**Figure 11.** Comparative study of $CO_2$ emissions between conventional and organically-farmed soils: (**a**) Farming system showing the four sampled plots; organic and conventional plots are sampled on the same day and at the same time. (**b**) Picarro G2201-I gas analyzer used in this study; and (**c**) $CO_2$ emissions (in ppm) measured in July and August 2018; first seven days were during dry soils, and last eleven days was during moist soils following rainfall. Results revealed that conventional farming emit more $CO_2$ than organically-farmed soils, and that soil respiration and emission are inhibited by high moisture content. Similar measurements of $CH_4$ were also included (not shown here).

## 4.3. Mine Tailing Dam Stability

Sustainable development and maintenance of infrastructure are vital when the social and economic impacts of failure are considered. We evaluate a new geophysical monitoring method to help define the stability and safety level of a mine tailing dam at Harmony Gold (Figure 12a). This should allow for early remedial maintenance and repair actions to be carried out that will improve public safety and reduce costs for owners, insurers and maintainers.

Seismic interferometry is used to image seepages and detect internal changes in the dam wall. The investigation consists of continuous ambient noise data recordings (at 400 Hz) over a period of 3 days with 20 triaxial short period geophones. The geophones are deployed over a survey wall of 100 m in length (Figure 12b). The horizontal components of the ambient noise data are cross-correlated, to approximate empirical green functions, and picked between sensor pairs to create surface wave dispersion curves (see 3.1). These dispersion curves are then inverted to estimate the shear wave velocity along the dam wall as a function of depth. We found that this method is successful in mapping the area of seepage along the cross-section. Monitoring over several weeks and in different seasons will require additional maintenance on the seismic stations (change on batteries, downloading of data, due to limited storage space and high sample rate) and will ultimately lead to improved interferometry of the tailings dam.

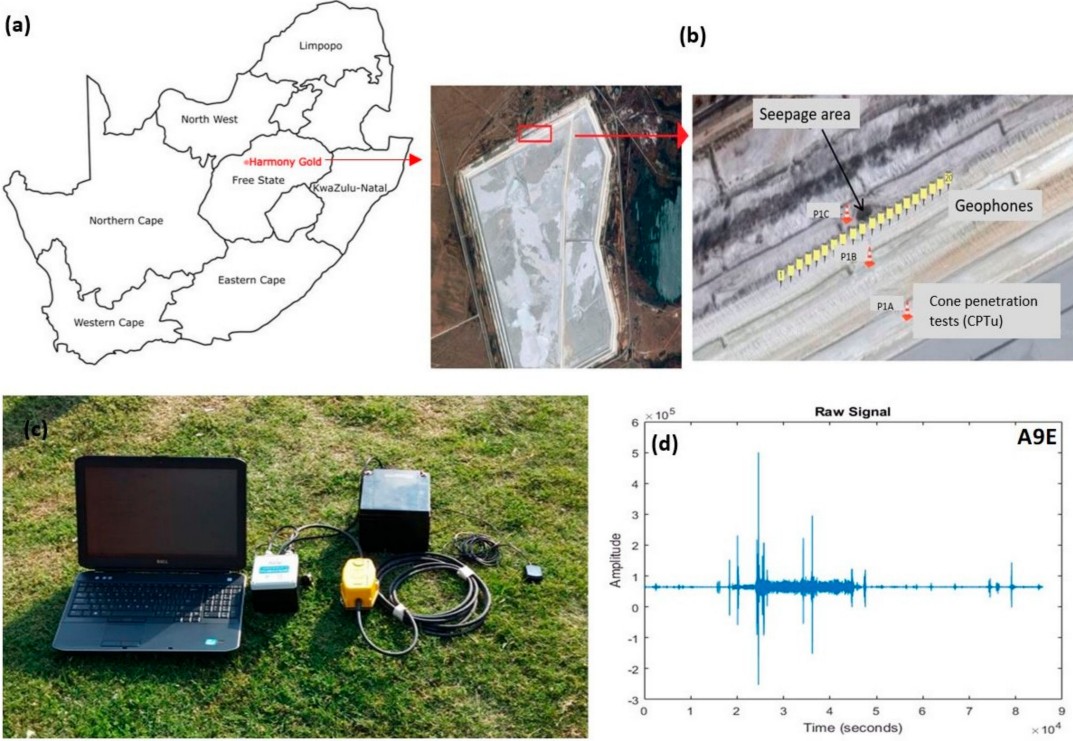

**Figure 12.** Passive seismic study of a mine tailing dam stability: (**a**) Location map of Harmony Gold; (**b**) aerial views showing the surveyed portion of the wall (red box) and the geophones positioned near an area where a seepage was detected; (**c**) instrumentation used to record the ambient noise; and (**d**) example of raw signal from one of these recording stations.

## 5. Citizen Science and Data Sharing

Citizen Science is an integral tool for community engagement in project design, implementation, data interpretation and reporting. This helps develop new knowledge, skills and support, as well as the building of a common AEON database.

### 5.1. Working Together with Local Communities

Our transdisciplinary research focusses on exploring the potential socio-economic impacts of extractive sectors (shale gas), which examines concepts, such as a social licence to operate and community agency, access to information, and public participation in public consultation processes. Deploying a mixed-methods approach in the research, this included household surveys in small towns of the Karoo, incorporating in-depth interviews and focus groups. Engagement with local leaders and the participation of community members are essential to this work and provide critical assistance in planning and ground truthing the municipal data and human settlement maps (Figure 13a), which are often outdated. This required further validation with the guidance and assistance of municipal officials and ward councillors. In conducting the surveys, local members of the community within each of the towns were trained and deployed as fieldworkers.

Experience gained from undertaking these surveys with the participation of local members of the community shows that communities become more aware of and alert to their rights, and engage more openly and freely with their local authorities through the existing structures. However, as technology advances and platforms, such as the internet become the dominant medium for communication and information sharing, and an enabler for more effective governance and citizenship the world over, poor communities with little access to such advancements may be left behind. Although spatial planning technologies are enhancing service delivery across the public sector, it is not clear as to how and when the Karoo rural municipalities will begin to see the benefits of such technologies,

both in terms of governance and citizenship, but also in terms of more effective service delivery to the inhabitants.

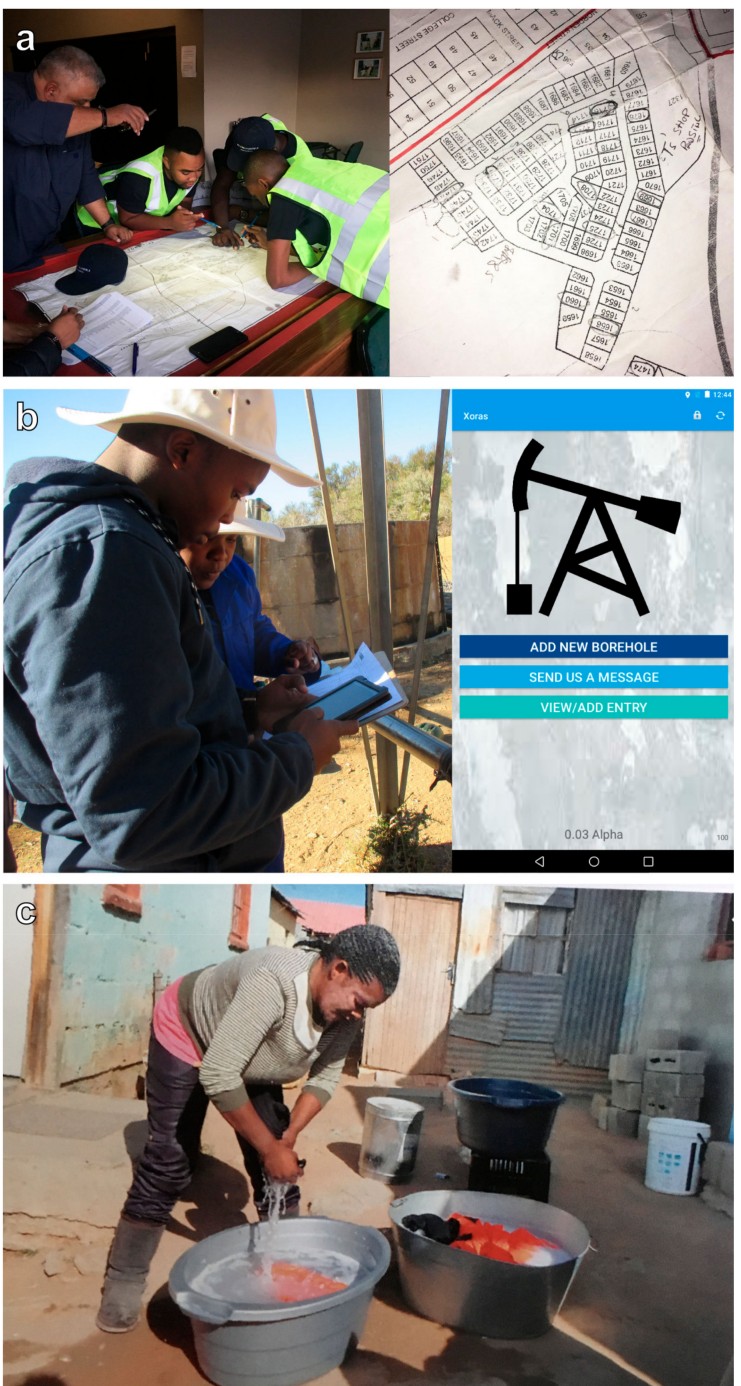

**Figure 13.** Citizen Science and household surveys in Karoo communities: (**a**) AEON researcher studying maps with fieldworkers of the housing department and a human settlement map in Jansenville; (**b**) hydrocensus trainees at a borehole site and the Xoras App interface; and (**c**) washing clothes using bucket systems in Lingilihle Township near Cradock. See Figure 1 for locations.

The understanding and application of Citizen Science during this study enhanced community engagement and facilitated the sampling of groundwater (Figure 13b). This consisted of iterative and participatory methodologies that include roundtable meetings, reflective journaling, focus group

discussions, as well as visual representations (photovoice methods) to establish, refine and prioritise community concerns.

To encourage sustainable management of water resources, we also evaluated how the local communities try saving water at a domestic level (Figure 13c). The focus was on the application of water conservation strategies in the household, while identifying possible influencing factors hindering or encouraging these actions in socio-economically different neighbourhoods. The survey revealed that water conservation is generally implemented across socio-economic boundaries, but that motivations and actions differ depending on financial income. Participants with low incomes often find themselves applying water-saving strategies without realising it (e.g., recycling water using bucket systems), while others with higher incomes actively inculcate equipment for, in particular, greywater harvesting.

Driven by the community's concerns over their lack of skills and knowledge capacity to monitor water quality in the studied region, the research included the training of young citizens, aged between 18 and 35 years, over a period of three years, from 2015 to 2018. The students were recruited using an inclusion criterion designed by a community representative group. An initial four weeks training on hydrocensus created an experiential learning environment. The training process was combined with the development of a customized 'Xoras' App to electronically capture and share the hydrocensus information from boreholes within the municipality (Figure 13b). The data were then validated and produced by AEON. This contributed to public knowledge about safe domestic water supply.

Social media platforms, such as WhatsApp speed up information sharing and allows us to communicate quickly with many people. We are testing this application to increase environmental awareness (in Hogsback; see 3.3). The aim is to convey an ecological viewpoint of wild animals, baboons in particular, as well as share data that can be used for the research. The participants provide day to day stories, and together with the scientific information, this increases the community knowledge.

*5.2. Sharing AEON Database Effectively*

AEON is building a database that includes all the data and metadata: Geological layers, ambient noise and animal recordings, hydrocensus, photographs, presentations, videos, and interviews collected during the different studies (Figure 14). The database is being expanded regularly with new and existing projects. It is divided into seven major fields, which then branch into speciality fields. Each transdisciplinary study has its own folder, subsequently branching to its subfolders to accommodate the raw data (measurements), processed data (analysis) and results (findings). A file with all the publications and academic thesis is also included.

Our GIS routinely integrates: Satellite and aerial images (from the national mapping agency), geosciences data from the Council for Geoscience (CGS); scanned and georeferenced maps; field stations and measurements. We combine this data to produce new maps and 3D models, particularly in using geophysical and borehole data (see 3.1). These maps and models are essential for data visualization and communication.

This system will be used to feed a new ESS data center for southernmost Africa, including an interface allowing for communication of lively information across all natural and human sciences to the public in form of newsletters, movies, virtual visits, interactive games, blogs, paintings, etc. (Figure 15). Object storage, adapted by some of the most successful private companies with huge data needs is view as a means to do this (e.g., Facebook, YouTube, Netflix, Google, Amazon). We would like to take this to Africa under 'Mandela Talks' [4]. This object storage for big data is needed to move from traditional filing systems to smartphones and be accessible to everyone.

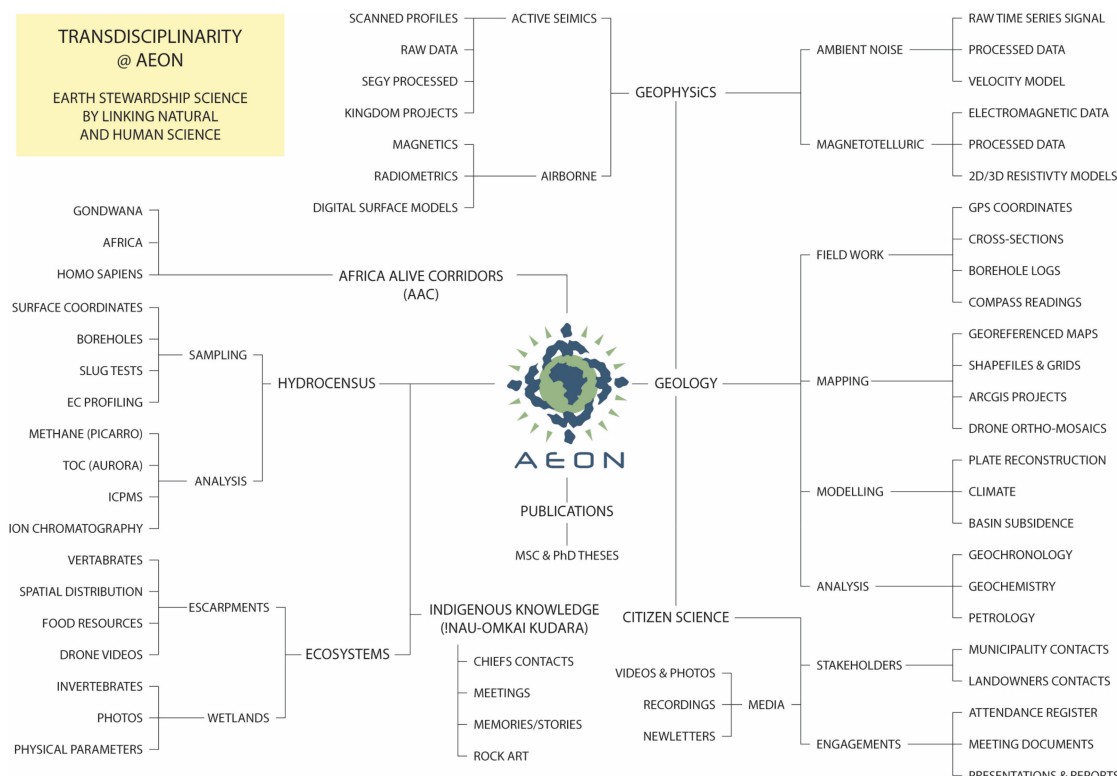

**Figure 14.** Database architecture used to manage AEON's earth stewardship science projects, publications and theses.

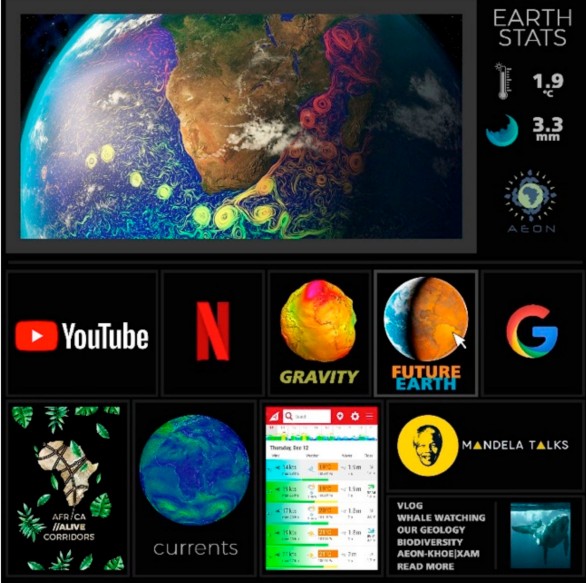

**Figure 15.** Prototype interface to communicate and share 'alive' earth and cultural data in southernmost Africa (see also 'Mandela Talks' [4]).

## 6. Discussion

Across southernmost Africa, *Homo sapiens* managed to live with and maintain endemic fauna and flora relatively intact over the course of nearly 200,000 years. At first, food security was generated by hunter-gatherers using wild plants (and first discovered rooibos tea), fish-traps (e.g., Figure 3a), primitive weapon and tool kits. However, following colonisation in the 17th century, human industries

rapidly replaced indigenous systems. As a result, in the last 100 years, a large proportion of the indigenous forest has been replaced with exotic plantations, farmlands, tourist game farms and extensive golfing estates across the studied region. Most of the wildlife is now restricted to game reserves, many of which are privately owned. In addition, marine life is being over-exploited and eradicated through plastic and shipping fuels. This has created an unacceptable disregard for the close relatives of early Khoisan who no longer have input into how to create and retain ecosystem services that existed before colonisation.

With population increase that will continue to double annually for more than 50 years, there is growing pressure on our natural heritage, both on living organisms, and with increasingly devastating consequences on the ability of our ecosystems to receive and hold adequate water resources [67]. Sustainability and planning ways of adapting to future climatic changes require transdisciplinary research, new ways of communication, data processing and preservation, and ESS as a fundamental teaching topic linking earth and human history.

*6.1. AEON's Transdisciplinary Research as a New Earth Stewardship Science*

Long-term monitoring is essential to preserve geo- and eco-systems and improve resources and climate risk management. As illustrated above, AEON projects integrate different disciplines and cover a range of subjects, including: Geological mapping, geophysics, hydrogeology, isotope chemistry, agriculture, various ecosystems, socio-economics, public engagement, politics and art. One major goal for all is to work together and commune with nature.

Better understanding the formation and evolution of the South African margin requires more field investigations and accurate dating techniques. The Earth's surface constantly evolves with erosion in the mountain regions and sedimentation along the shelf and in deep oceanic basins. Numerous tectonic and climatic processes operate concurrently at different spatial and temporal scales, from the supercontinent (Wilson) cycle of 250 million years to the season and day (e.g., tide, El Niño, Whakaari), which implies broad and integrated land-sea research approaches. One major challenge is monitoring vertical movements of the continent and distinguishing these from global sea-level changes. Our work clearly shows that a unique, long and diverse history of changes is best preserved along the coastal region (Figure 4). We propose enhancing field schools and geophysical experiments, both inland and offshore, to help teaching young earth stewardship scientists, as well as to increase the resolution of our current tectonic and climate models. This research must utilise drone technologies to improve the conventional measurements and to help predict future land and sea-level rises, and coastal erosion, which are all linked to the increased frequency of thunderstorms, tsunamis, and the predicted 2–4 °C global warming by 2100 [68].

Seismic (and sound) instrumentation is becoming more precise and easier to deploy, and more fundamental research in this field is required. We have shown that passive methods can be used to survey subsurface geology and groundwater, as well as to monitor the stability of infrastructure (Figure 12). Such techniques have many other applications in areas yet to be investigated.

Remote acoustic sensing can be used to detect changes in all species' lifestyles, including humans, linked to climate and anthropogenic changes. This may be easier on land than at sea. In Algoa Bay (Figure 5), for example, it is limited to point sampling, which prevents the extrapolation of data to a larger geographic area. Assessing the contribution of human-generated noise by large ships to the marine soundscape, requires modelling of sound propagation from the location of known sources. However, this is often challenging because of lack of high resolution maps of the bathymetry, temperature, salinity and geology, and how these change with depth. Through this work, we have realised how much noise we make in the ocean and that this affects virtually all marine species, but more work is needed to detect and anticipate future changes.

Long term monitoring of groundwater-dependent ecosystems is essential for the collection of a required dataset to understand the groundwater dynamics. This allows for more accurate modelling and prediction of contamination and excessive groundwater abstraction that is usually exacerbated by

droughts as predicted in South Africa [69,70]. We started these baseline studies in the Eastern Cape, in anticipation of potential shale gas extraction [71]. The goal was to establish pre-drilling conditions in advance of and to avoid adverse environmental and social legacies, as have been encountered elsewhere in the mining industry. Different sources of salinity that are critical to better understand and precisely map for an increasing supply of freshwater to rural communities and large urban centres developing near the coast have been discovered (Figure 9). Drone technologies are now required for effective surveys; with on-board geophysical methods, these aerial surveys could map the groundwater tables and help detect new sources.

Across the Karoo, shallow groundwater has the potential to contain relatively high U concentrations, which lead to a geo-health risk on humans and animals. For monitoring of these geo-medical contaminations, community engagement, such as the inquiry into clinical data showing statistics and incidence of diseases is needed. The application of Citizen Science in working with rural communities had significant benefits for the process of building an informed and critical citizenry, which is empowered with first-hand knowledge on resources and future developments in their communities. This envisioned future outcome is sorely needed today, as we must make the transition to a more sustainable future. In this transition, ordinary people can no longer rely on their elected leaders to create the future that is needed as poor governance and unaccountable leadership is glaringly evident today. Instead, communities and social formations must begin to build networks of influence and action that will drive existing institutions towards the desired end state. Deploying this strategy at the local and regional scale, through building citizen science networks and collaborative programmes between universities and communities is essential; and this must be expanded nationally and globally.

## 6.2. Telling Stories for and by Everyone—Africa Alive Corridors

Cataloguing our natural and cultural heritage and making this information available is an important step in determining what we need to sustain Earth as a habitable planet. The Africa Alive Corridors (AAC) program of AEON, a spin off from a larger global program named 'Gondwana Alive' [72,73], is designed to be a two-way conduit for knowledge transfer between academic institutions and African communities (Figure 16).

The AAC project aims at telling and celebrating Africa's four billion-year heritage that entails its geological, biological and cultural evolution. The convergence between the sciences and humanities are critical in the quest to achieve such goals. The development of the AAC through transdisciplinary sciences implies new processes of learning, towards earth stewardship for civil society, universality, and redefining the values that govern human and animal well-being. Finding itself at the science-society interface, the AAC will ultimately communicate its work to communities across Africa. Multilingual datasets and visualisations are important tools to transmit this information. Through aesthetic map construction and participatory mapping, AEON is producing integrated information about the rich biodiversity and its link to geological processes, and the evolving Anthropocene, such as described in this study.

In southernmost Africa, the *Homo sapiens* corridor tells the exceptional story of the first half of our human journey on Earth, from about 140,000 to 60,000 years ago. The proposed roadmap provides the best geological, archaeological and biological records from which to explore this history, and develop a more responsible earth steward attitude to protect our natural and cultural heritage for future generations.

Having started more than 20 years ago with the Gondwana Alive project, the general idea is to promote biodiversity and to try to stem the tide of mass extinction and global ecological catastrophes [74]. These ideals were originally endorsed by Nelson Mandela in 1998: "Because it approaches, as I see it, the very core of two concerns most dear to me – the children of today's world and the children of tomorrow's world". The process of encouraging the participation of all Africans was then taken to the launch of the International Year of Planet Earth (IYPE) in May 2008 in Arusha, Tanzania, where young scholars from South Africa and Tanzania played 'the AAC card game' to protect the African continent

against population growth and global warming [75]. Today, it is vital to make the stories of Africa 'alive'. Our southern Africa project is an example that we believe should be taken further across Africa and Gondwana.

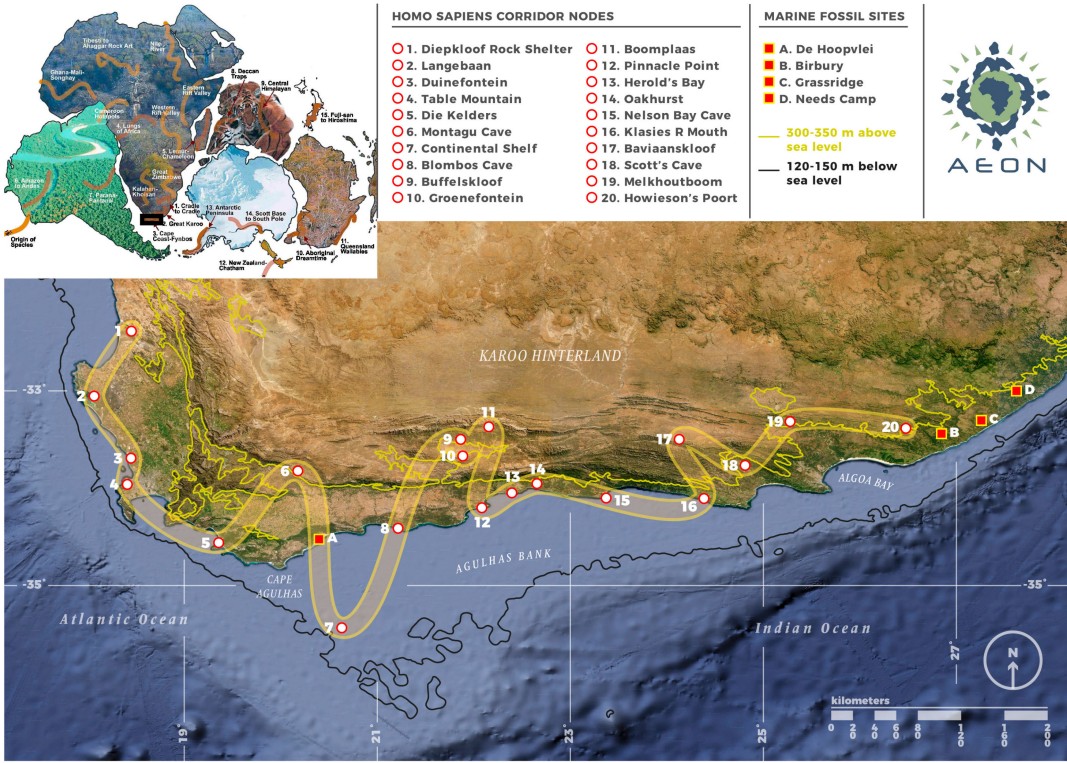

**Figure 16.** *Homo sapiens* corridor showing important natural and cultural heritage nodes; each of these sites is to be covered by geological and biological mapping using drones and smartphones. Inset shows its location in the Gondwana Alive Corridors (GAC) framework (black box) [73]. Note this is AAC 10, one of the 20 corridors across Africa, presently under final construction [2].

## 7. Conclusions

This paper aimed to highlight some of the interactive research consideration toward establishing efficient and effective earth stewardship science research toward controlling an ever-evolving natural and societal landscape. Nelson Mandela University (home of AEON) is in a nature reserve and marine protected area along the southern coast of South Africa. This area has rich and interactive marine and terrestrial systems that provide a unique natural laboratory for transdisciplinary, earth stewardship research. Our goal is to build an all-encompassing and dynamic database that is accessible to all and sheds light on past, present and possible future evolution. It should serve as a standard to document, monitor and communicate sciences across other regions in Africa and elsewhere, through a new way of learning ESS.

This remains an evolving science, and several pertinent overarching questions still require answering: e.g., how does the natural environment and Earth's entire history inform humanities to make effective decisions to address the future? What natural response can be expected from the Earth? AEON is continuously evolving earth stewardship research methods to better understand the complex evolution of Africa, particularly along its southernmost coastal region, with the goal of providing a means to improve society relationship with their natural environment. This work also aims to develop the necessary tools and technologies that can support important industrial and socio-economic development in the most sustainable manner.

To integrate various fields and build interconnectivities as a remote sensing study for cross-disciplinary research, or more appropriately named Earth Stewardship Science is, to say the

least, challenging. This collaborative effort is critical for documenting and monitoring our natural and cultural heritage, and should serve as a basis for future research requiring integration of geospatial data at different scales with the social and economic factors. Making knowledge accessible through new curricula, education programs and inclusion in projects will give local communities the opportunity to be better custodians of their natural environment.

**Author Contributions:** Conceptualization, B.L.; investigation, K.W., R.P., M.M., K.V., L.B., T.D., T.M., V.N., T.N., D.C., T.K., T.S., R.S., S.M., L.M., F.G., G.G., M.B., P.M., N.K., K.J., S.D., O.P., G.M., J.B., N.G., D.S., B.M., N.D., V.S., C.R., T.T., W.M. All authors have read and agreed to the published version of the manuscript.

**Funding:** This research was funded by the DST/NRF through Iphakade. This AEON's publication number 196 and Iphakade publication number 241.

**Acknowledgments:** This work was inspired and encouraged through several of AEON's original 15 founder members, in 2006, all from different academic backgrounds: Christien Thiart (Statistics), Judith Masters (Biology), Moctar Doucouré (Geophysics), Maarten de Wit (Geology) and John Anderson (Zoology and Palaeontology); as well as new members: Gavin Bradshaw (Politics and Conflict Resolution) and David Jones (Fine Arts) from Nelson Mandela University. We also thank three anonymous reviewers for their constructive comments and suggestions.

**Conflicts of Interest:** The authors declare no conflict of interest.

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
