# Peer review of "Earth Stewardship Science—Transdisciplinary Contributions to Quantifying Natural and Cultural Heritage of Southernmost Africa"

_remotesensing, doi:10.3390/rs12030420_

Round 1

Reviewer 1 Report

The manuscript shows many applicative fields in Earth Stewardship science. The topic is very interested and useful.

The reviewer suggests that this paper should focus on quantifying methods or experiences on how to conduct to natural and cultural heritage of southernmost Africa, but not to give a general introduction to those research fruits over the past years. The authors could single out a few outstanding examples or cases to present the “Transdisciplinary ” and “quantifying” research, such as the manuscript’s “models integrate offshore seismic and borehole data” . The authors could present a brief summary of AEON’s contributions in another independent section at the paper’s last part. Those structure of the paper may be more vivid and not boring to read.

Author Response

The reviewer’s comments are positive and we took them into account during correction, however we did not change the entire structure of the paper as proposed. We believe that it is important to keep the detailed descriptions of all the different methods and data acquired during this study in order to support our transdisciplinary approach and the results.

In section 3, we have added some missing details on the quantifying methods:

3.1. Geological and geophysical mapping. Here we have detailed the satellite and drone image processing method. We have added some information and results on our models that integrate offshore seismic and borehole data. We have completed the description of the passive seismic data processing.

3.2. Listening-to and timing-off evolving marine ecosystems. Here we have detailed the method used for the marine soundscape monitoring and added some references to recent works.

Reviewer 2 Report

Dear Editor ,
Journal: Remote Sensing (ISSN 2072-4292)

Manuscript ID: remote sensing-688471

Title: Earth Stewardship Science - Transdisciplinary contributions to quantifying natural and cultural heritage of southernmost Africa

The manuscript has good scientific result on Transdisciplinary contributions to quantifying natural and cultural heritage of southernmost Africa. The paper may be considered for publication in Journal of remote sensing  after major corrections.
Comments to Author:
General Remarks: 

1)Resolution of figures, is poor and labels ( Figure 3,5, 6,16)

2) Figures 12 and 13 : decrease the photos of fieldworkers and training members of the local community (one or two photos are enough in the scientific paper )

3)The references must be prepared using instructions for authors.

Specific:

1) The ‘Abstract’ must be revised and improved, in order to be better correlated with the content of the paper and objectives. Please consider that the results should represent the most important part of the ‘Abstract’. You should address briefly the obtained results.

Introduction Section:

According to the aim of the paper: to establish what we can learn from the past short- and long-term history about precipitation and sea level changes, and their variabilities with time. Some of these questions can be answered better than anywhere else in (South) Africa, provided that the data is quantified collectively. Here we demonstrate how this region can be used as an earth-humanities baseline against which to measure and predict changes into the future. I suggest to change the title of this article .

Experimental Section:
1)What does “Sclerochronology’’ mean ?  more definitions ?
 Results and Discussions Section:

1)Line 498 : The identification of Seismic interferometry is used to image see pages and detect internal changes in the dam wall, carried out by which software and a description the method ?

2)Lines 171 and 172 – sentences need editing

3)Lines 706 and 707 the children of today's world and the children of tomorrow’s world” sentences need changes.

4) The text last paragraph‘6.2. Line 682 to 697 should added some others references .

Exemples

Anderson, J.M., de Wit, M.J. & van Heerden, A. 2008. Earth Alive; 101 strategies towards stemming the Sixth Extinction & global warming. Minimag, Pretoria, 27 pp (this was first published by Minimag, a magazine for South African schoolgoers, as a 10-part monthly series through 2007).

Anderson, J.M. & De Wit, M.J., 2005. Gondwana Alive: at the threshold, Abstract, 12th International Gondwana Symposium, Mendoza, Argentina, November 2005.

......

......

Finally, I would like to congratulate the authors for the effort and scope of the article. It presents an interesting method/topic and has high readability and interest to readers. Regarding the manuscript, the current form needs minor-medium revisions. Some aspects need to improved (ex. abstract, introduction). The overall English and spelling are very good.

Author Response

Our answer are in highlighted in red below:

General Remarks: 

1)Resolution of figures, is poor and labels ( Figure 3,5, 6,16)

2) Figures 12 and 13 : decrease the photos of fieldworkers and training members of the local community (one or two photos are enough in the scientific paper )

3) The references must be prepared using instructions for authors.

In the new version of the paper, we have 1) inserted all the figures in high resolution and revised the label. 2) We have combined Figures 12 and 13 into one; and 3) we have corrected the reference list according to the Remote sensing standards.

Specific:

1) The ‘Abstract’ must be revised and improved, in order to be better correlated with the content of the paper and objectives. Please consider that the results should represent the most important part of the ‘Abstract’. You should address briefly the obtained results.

The reviewer is correct, and we have added a brief description of the results in our abstract, as follow:

“We find that the history of this margin is highly episodic and complex by, for example, the successful application of ambient noise and groundwater monitoring to assess the evolving human impacted ecosystems. This is being explored also with local Khoisan representatives and rural communities through Citizen Science.”

Introduction Section:

According to the aim of the paper: to establish what we can learn from the past short- and long-term history about precipitation and sea level changes, and their variabilities with time. Some of these questions can be answered better than anywhere else in (South) Africa, provided that the data is quantified collectively. Here we demonstrate how this region can be used as an earth-humanities baseline against which to measure and predict changes into the future. I suggest to change the title of this article.

We did not change the title of our article. Following the reviewer’s comment, we have improved the description of our aim, as follow:

“Our aim is to define and produce transdisciplinary knowledge of southern Africa’s natural and cultural heritage, and package it in a way that is widely accessible, effectively producing Earth Stewards out of local communities and students.”

Experimental Section:
1) What does “Sclerochronology’’ mean ?  more definitions ?

We have re-written the definition: “Sclerochronology [43,44], which is the study of periodicities preserved in growth patterns of layered biogenic carbonates, such as stromatolites and bivalve shells (Figures 5d,e), is also an ideal quantitative tool to reconstruct the environmental changes across Algoa Bay. Using geochemistry, we can translate the growth and temporal record of the organism into proxies for physical parameters such as seasonality, tides and salinity.”

Results and Discussions Section:

1) Line 498 : The identification of Seismic interferometry is used to image see pages and detect internal changes in the dam wall, carried out by which software and a description the method ?

We have described the method and software used for the passive seismic as requested by the reviewer in section 3.1:

“Prior to pre-processing, the continuous ambient seismic noise waveform data from the triaxial components are converted from Cube to Miniseed format and divided into daily (24 hour) segments. Additional format conversion for waveform analysis are done in Matlab, Msnoise and Msnoise-TOMO. Pre-processing is automated and completed for each station to prepare the waveforms for correlation. After cross-correlation between all station pair’s, dispersion measurements and tomographic inversion are performed. The local dispersion measurements are jointly inverted to create The results are isotropic shear velocity (vs) models”

In section 4.3 , we have improved the text, as follow:

“Seismic interferometry is used to image seepages and detect internal changes in the dam wall. The investigation consists of continuous ambient noise data recordings (at 400 Hz) over a period of 3 days with 20 triaxial short period geophones. The geophones are deployed over a survey wall of roughly 100 m in length (Figure 12b). The horizontal components of the ambient noise data are cross-correlated, to approximate empirical green functions, and picked between sensor pairs to create surface wave dispersion curves (see 3.1).”

2) Lines 171 and 172 – sentences need editing

We have re-written the sentences as follow:

“The supply and distribution of potable water is one of the major challenges currently facing the communities across the studied region, as illustrated by the onset of the 2015 drought in Cape Town. By the end of the dry season in May 2017, the drought was declared the city’s worst in a century [27-25]. Then in February 2018 and October 2019, many other towns were declared drought disaster areas following the several years of precipitation well below average rainfall; e.g. Port Elizabeth received 42 % below the average amount of 271 mm between January and July in 2018 [28].”

3) Lines 706 and 707 the children of today's world and the children of tomorrow’s world” sentences need changes.

We cannot change this citation as these are the exact words used by Nelson Mandela. We therefore do not understand the issue of this review.

4) The text last paragraph‘6.2. Line 682 to 697 should added some others references .

Exemples

Anderson, J.M., de Wit, M.J. & van Heerden, A. 2008. Earth Alive; 101 strategies towards stemming the Sixth Extinction & global warming. Minimag, Pretoria, 27 pp (this was first published by Minimag, a magazine for South African schoolgoers, as a 10-part monthly series through 2007).

Anderson, J.M. & De Wit, M.J., 2005. Gondwana Alive: at the threshold, Abstract, 12th International Gondwana Symposium, Mendoza, Argentina, November 2005.

We have added these 2 references to the text in section 6.2 and in the reference list.

Finally, I would like to congratulate the authors for the effort and scope of the article. It presents an interesting method/topic and has high readability and interest to readers. Regarding the manuscript, the current form needs minor-medium revisions. Some aspects need to improved (ex. abstract, introduction). The overall English and spelling are very good.

As described above we have improved the abstract and introduction, after which we have added 2 new paragraphs on landscape archaeology (see Review 3).

Reviewer 3 Report

The research presents an interesting development for the study and analysis of the evolution of human impact on the region. The multidisciplinary approach is the strength of the research and I would deepen, where possible, the study of landscape archeology. Through the spatial analysis tools provided by GIS, integrated with the applications of algorithms for the reconstruction of the landscapes used in satellite Remote Sensing, it's possible to outline the development of the evolution of the landscape and the human role, with the acceleration in the modern era, like programs Iphakade: ‘Observe the present and consider the past to ponder the future’. The article has great potential and a good structure, but I think it is important to present the data of the initial collection on cultural heritage, as anticipated in 2.2. Cultural Heritage: ‘first come-last served’. This would make the structure stronger and would allow the approach to the study of the evolution of the landscape from an ancient human perspective, linked to the interaction and respect for nature.

Author Response

The reviewer’s comments are valid and very useful. In section 2.1 we have added descriptions of the archeological and cultural heritage. As follow:

“Landscape archeology that is based on evidence of early human habitation is unique across the studied region (Figures 1 and 2). Here, the earliest evidences of our cultural evolution have been re-discovered, including: classic Acheulian stone tools (e.g. Montagu Cave, 500-200 ka); the oldest known engraved rock and some necklace beads made from gastropod shells (Blombos Cave, 75 ka); a variety of ostrich eggshell containers (Diepkloof Rock Shelter, 65-55 ka) [12-13]; and the earliest human footprints along sand dunes (Langebaan, 117 ka) [14]. These, together with dated fossil records, reveal strong relationships between human evolution, sea-level and climate changes, with the most important modifications occurring during intervals of maximum glaciation (‘ice ages’), such as the Cape coast is envisaged to represent the cradle of human culture.

Around Cape Town, there are more tragic archives of the recent stories of how the first European colonizers removed, killed and incorporated the indigenous San hunter-gatherers and the Khoi pastoralists as slaves [15]. This knowledge has, for example motivated the Khoisan to advocate the reburial of Sschura (Sarah Baartman in 2001) in the Eastern Cape. This ‘First Nation’ is re-experiencing their spiritual connectiveness to the past, and which compels current chiefs and activists to seek restorative justice for the loss of identity, tradition and land, based on archeology and landscapes that can now be carefully re-mapped using drone technology (Figure 2).”

In addition, to better describe the landscape archaeology, we have added a new Figure 2: “Aerial drone photo of a cave eroded in synclinal folds of the Table Mountain Group quartzites, bordering the Tsitsikamma indigenous forest along the southern coast (Figure 1 for location). The flat erosion surface elevated at more than 200 m above sea level is a wave-cut platform that record southern Africa uplift and fluctuated global sea-level drop since about 100 Ma.”

Round 2

Reviewer 2 Report

Dear Authors,
you made a great effort in responding to all questions arose from the first step of the review process.
From my side, your manuscript is ready for publishing in its present form.